# Seeing the Unseen: How EMoE Unveils Bias in Text-to-Image Diffusion Models

## Abstract

Estimating uncertainty in text-to-image diffusion models is challenging due to their large parameter counts (often exceeding 100 million) and operation in complex, high-dimensional spaces with virtually infinite input possibilities. In this paper, we propose Epistemic Mixture of Experts (EMoE), a novel framework for efficiently estimating epistemic uncertainty in diffusion models. EMoE leverages pre-trained networks without requiring additional training, enabling direct uncertainty estimation from a prompt. We introduce a novel latent space within the diffusion process that captures model uncertainty better during the first denoising step than existing methods. Experimental results on the COCO dataset demonstrate EMoE's effectiveness, showing a strong correlation between uncertainty and image quality. Additionally, EMoE identifies under-sampled languages and regions with higher uncertainty, revealing hidden biases related to linguistic representation. This capability demonstrates the relevance of EMoE as a tool for addressing fairness and accountability in AI-generated content.

## 1 Introduction

In recent years, text-to-image diffusion models have made remarkable strides, enabling faster image generation (Song et al., 2020; Liu et al., 2023; Yin et al., 2024), improved image quality (Dhariwal & Nichol, 2021; Nichol et al., 2022; Rombach et al., 2022), and even extending into video generation (Ho et al., 2022b; Khachatryan et al., 2023; Bar-Tal et al., 2024). Diffusion models operate through a two-phase process: in the forward phase, noise is incrementally added to the data, while in the reverse phase, the model learns to denoise and reconstruct the image. However, despite their growing popularity, these models often function as black boxes, providing little transparency into their decision-making processes or how they handle uncertainty (Berry et al., 2024; Chan et al., 2024). To address these limitations, we introduce Epistemic Mixture of Experts (EMoE), a novel framework for capturing and quantifying epistemic uncertainty in text-conditioned mixture-of-experts diffusion models, which are capable of generating high-resolution images ($512 \times 512 \times 3$). Epistemic uncertainty, arising from a model's lack of knowledge, can be reduced with additional data, whereas aleatoric uncertainty, stemming from inherent randomness in the data, remains irreducible (Hora, 1996; Der Kiureghian & Ditlevsen, 2009; Hüllermeier & Waegeman, 2021).

An example of our approach is illustrated in Figure 1. The top row contains images for the prompt, "A white man holding the office of the US President" with low epistemic uncertainty (0.32), followed by "A black man holding the office of the US President" with an uncertainty 0.34. The bottom row displays images for the prompt, "A white woman holding the office of the US President" with an uncertainty 0.43, followed by "A black woman holding the office of the US President" with high epistemic uncertainty (0.60). This comparison highlights potential biases in the model's handling of demographic diversity across race and gender. To our knowledge, EMoE is the first framework to effectively capture epistemic uncertainty in text-conditioned diffusion models for image generation.

The EMoE framework is built on two key components. First, it leverages pre-trained mixture-of-experts (MoE) for zero-shot uncertainty estimation. Notably, the experts in the MoE were not trained for uncertainty estimation but were independently trained on different datasets. Originally introduced by Jacobs et al. (1991), Mixture-of-Experts (MoE) models form ensembles in sub-modules, where each expert specializes in specific tasks, benefiting from a shared base model to ensure efficiency while harnessing the collective power of multiple experts (Shazeer et al., 2017). Training

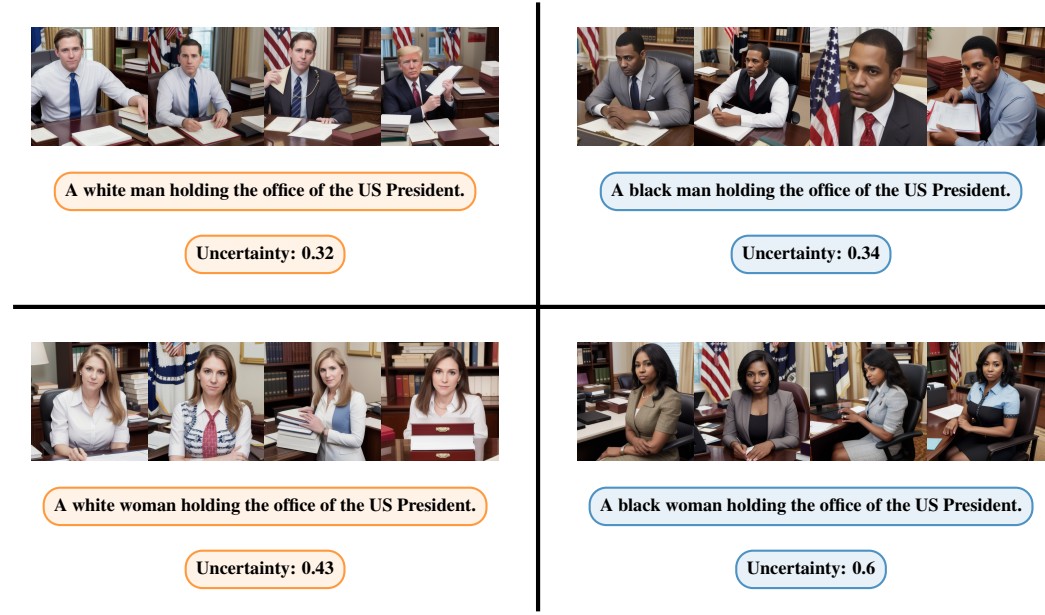

Figure 1: This figure illustrates the uncertainty levels for different demographic prompts related to the US President. The model demonstrates the lowest uncertainty (0.32) for a white male president, followed by a black male president (0.34) and a white female president (0.43). The highest uncertainty (0.6) is observed for a black female president, highlighting potential biases in the model's handling of demographic diversity in race and sex.

such an ensemble of diffusion models from scratch is computationally expensive, requiring hundreds of GPU-days on current hardware (e.g. Nvidia A100 GPUs) (Balaji et al., 2022). By leveraging pre-trained ensembles, EMoE achieves substantial computational savings and applies MoE in a novel context.

The second key component of the EMoE framework is that it estimates uncertainty on a novel latent space identified by probing the intermediate activations of the diffusion model's denoiser. This space enables the model to identify regions in the input space (i.e., prompts) where hallucinations or incorrect image generation are more likely. By disentangling the expert ensemble components and measuring variance within this space, EMoE can detect high epistemic uncertainty early in the denoising process and thereby offer a more proactive assessment than previous methods that evaluate uncertainty after image generation (Song et al., 2024).

By combining pre-trained experts with a novel latent space for uncertainty estimation, EMoE addresses the challenge of quantifying epistemic uncertainty in text-conditioned diffusion models. We evaluate EMoE's performance on the Common Objects in Context (COCO) dataset Lin et al. (2014), and our contributions are as follows:

- We establish the EMoE framework for text-conditioned diffusion models, leveraging pre-trained experts and introducing uncertainty estimation within a novel latent space in the network (Section 3).
- We demonstrate the effectiveness of EMoE for image generation on the COCO dataset, a widely used and challenging benchmark, and show that EMoE aligns with expectations of epistemic uncertainty (Section 4.1).
- We further evaluate EMoE's ability to detect novel data by assessing which languages the model has previously encountered and examining the bias inherent in diffusion models. This analysis is conducted across 25 different languages (Section 4.2 & Section 4.3).
- We justify our design choices by conducting a set of ablation studies (Section 4.4).

These contributions shed new light on the previously opaque area of epistemic uncertainty in text-conditioned diffusion models, offering significant implications for risk assessment and decision-making processes in sensitive domains.

## 2 BACKGROUND

Diffusion models construct a Markov chain, where each step involves sampling from a Gaussian distribution. This setup is well-suited for uncertainty estimation, as probability distributions naturally lend themselves to uncertainty reasoning (Hüllermeier & Waegeman, 2021). Furthermore, MoE models are particularly effective at capturing epistemic uncertainty, as they leverage an ensemble of experts, which can be viewed as a Bayesian approximation (Hoffmann & Elster, 2021).

### 2.1 DIFFUSION MODELS

In the context of supervised learning, consider a tuple $(x, y)$, where $x$ represents an image of size $512 \times 512 \times 3$ and $y$ is the prompt associated with the image. The objective is to estimate the conditional distribution $p(x|y)$, which is challenging due to its high-dimensional, continuous, and multi-modal nature. In this work, we use latent diffusion models (Rombach et al., 2022), a powerful model for arbitrary data distributions which reduces computational costs by operating in a latent space learned by an autoencoder. The autoencoder consists of an encoder $\mathcal{E}$, which maps images to their latent representation, and a decoder $\mathcal{D}$, which does the opposite.

Diffusion models use a two-phase approach, consisting of a forward and a reverse process, to generate realistic images. In the forward phase, an initial image $x$ is encoded to $z_0$ and then gradually corrupted by adding Gaussian noise over $T$ steps, resulting in a sequence of noisy latent states $z_1, z_2, \ldots, z_T$. This process can be expressed as:

$$q(z_t|z_{t-1}) = \mathcal{N}(z_t; \sqrt{1 - \beta_t} z_{t-1}, \beta_t \mathbf{I}) \qquad q(z_{1:T}|z_0) = \prod_{t=1}^{T} q(z_t|z_{t-1}), \tag{1}$$

where $\beta_t \in (0, 1)$, with $\beta_1 < \beta_2 < \cdots < \beta_T$. This forward process draws inspiration from non-equilibrium statistical physics (Sohl-Dickstein et al., 2015).

The reverse phase of the process aims to remove the noise and recover the original image, conditioned on text. This is achieved by estimating the conditional distribution $q(z_{t-1}|z_t, y)$ through a model $p_\theta$. The reverse process is defined as:

$$p_\theta(z_{0:T}|y) = p(z_T) \prod_{t=1}^{T} p_\theta(z_{t-1}|z_t, y) \qquad p_\theta(z_{t-1}|z_t, y) = \mathcal{N}(z_{t-1}; \mu_\theta(z_t, t, y), \Sigma_t). \tag{2}$$

where $p_\theta(z_{t-1}|z_t, y)$ represents the denoising distribution, parameterized by $\theta$, and is modeled as a Gaussian with mean $\mu_\theta(z_t, t, y)$ and covariance $\Sigma_t$. While $\mu_\theta$ is an output of the learned model, $\Sigma_t$ follows a predefined schedule, such that $\Sigma_0 < \Sigma_1 < \cdots < \Sigma_T$. These forward and reverse processes together form a Markov chain, driving the image generation.

Given the complexity of directly computing the exact log-likelihood $\log(p_\theta(z_0|y))$ in the reverse process, it is common to use the Evidence Lower Bound (ELBO) (Kingma & Welling, 2013) as a tractable surrogate objective. The ELBO provides a lower bound on the log-likelihood and can be expressed as:

$$-\log(p_\theta(z_0|y)) \leq -\log(p_\theta(z_0|y)) + D_{KL}(q(z_{1:T}|z_0) \parallel p_\theta(z_{1:T}|z_0, y)). \tag{3}$$

where the goal is to balance two terms: maximizing the likelihood of the original image $z_0$ and minimizing the Kullback-Leibler (KL) divergence between the true posterior distribution $q(z_{1:T}|z_0)$ and the approximate posterior $p_\theta(z_{1:T}|z_0, y)$. Using properties of diffusion models, this ELBO formulation leads to a specific loss function that optimizes the noise-prediction model:

$$L_{LDM} = \mathbb{E}_{z, \epsilon \sim \mathcal{N}(0,1), t, y} \left[ ||\epsilon - \epsilon_\theta(z_t, t, y)||_2^2 \right]. \tag{4}$$

where $t$ is uniformly distributed over $1, ..., T$, $\epsilon \sim \mathcal{N}(0, 1)$, and $\epsilon_\theta(z_t, t, y)$ is the predicted noise for computing $\mu_\theta(z_t, t, y)$. For details, see Ho et al. (2020).

### 2.2 U-NETWORKS

U-Nets, a Convolutional Neural Network (CNN) architecture originally developed for biomedical segmentation, have demonstrated their effectiveness across a range of generative tasks, including

image synthesis and restoration (Ronneberger et al., 2015; Isola et al., 2017). Their encoder-decoder structure is well-suited for pixel-level predictions, as it captures both global context and fine details.

A U-Net consists of a downsampling path (i.e. $\mathrm{down}$-blocks), an upsampling path (i.e. $\mathrm{up}$-blocks), and a mid-block. The downsampling path compresses the input $z_t$ into a latent representation $m_t^{\mathrm{pre}}$, where $\mathrm{down}(z_t) = m_t^{\mathrm{pre}}$, by reducing spatial dimensions and increasing the number of feature channels. The $\mathrm{mid}$-block refines this latent representation into $m_t^{\mathrm{post}}$, where $\mathrm{mid}(m_t^{\mathrm{pre}}) = m_t^{\mathrm{post}}$. The up-block then reconstructs the image by upsampling $m^{\mathrm{post}}t$ to $zt-1$, the next latent representation in the denoising process. This process effectively combines low-level details with high-level semantic information.

U-Nets are widely used in diffusion-based generative models, where they model $\epsilon_\theta(z_t, t, y)$, effectively removing noise while preserving structure. The ability to maintain both local and global information through skip connections makes U-Nets particularly suited for diffusion models.

To then make our models conditional on a prompt $y$, we map $y$ through a tokenizer $\tau_\theta$ and pass this intermediate representation within the $\mathrm{down}$-, $\mathrm{mid}$- and $\mathrm{up}$- blocks via a cross-attention layer Attention$(Q, K, V) = \mathrm{softmax}\left(\frac{QK^T}{\sqrt{d}}\right) V$ (Vaswani et al., 2017). We mathematically denote this as follows:

$$Q = W_Q \phi_\theta(z_t) \qquad K = W_K \tau_\theta(y) \qquad V = W_V \tau_\theta(y). \tag{5}$$

Here, $W_Q$, $W_K$, and $W_V$ are learned projection matrices, and $\phi_\theta(z_t)$ and $\tau_\theta(y)$ represent the encoded latent representations of the inputs $z_t$ and tokenized input $y$. The cross-attention output is then passed through a feed-forward neural network, as in the transformer architecture.

## 2.3 Sparse Mixture of Experts

MoE is a widely-used machine learning architecture designed to handle complex tasks by combining the outputs of several specialized models, or "experts" (Jacobs et al., 1991; Shazeer et al., 2017). The key intuition behind MoE is that different experts can excel at solving specific parts of a problem, and by dynamically selecting or weighing their contributions, the overall model can perform more effectively. MoE models are particularly useful in cases where the data is heterogeneous, involving a variety of sub-tasks or domains that benefit from expert specialization.

MoE combines multiple expert models by forming an ensemble, utilizing cross-attention layers and feed-forward networks embedded within the U-Net architecture. Let $M$ denote the number of experts, and let $i$ denote the $i$-th expert. The cross-attention layer can then be expressed as:

$$Q^i = W_Q^i \phi_\theta(z_t), \qquad K^i = W_K^i \tau_\theta(y), \qquad V^i = W_V^i \tau_\theta(y). \tag{6}$$

The matrices $W_Q^i$, $W_K^i$, and $W_V^i$ are learned projection matrices specific to each expert $i$, allowing each expert to attend to different aspects of the input information.

A similar process occurs within the feed-forward networks, where each expert processes the data independently before their results are combined (Lepikhin et al., 2020). The ensemble created by this mechanism leads to more robust predictions, as each expert is able to specialize and contribute uniquely to the final output. In addition to the ensemble created by the cross-attention and feed-forward layers, the MoE architecture includes a routing or gating network that dynamically selects which experts to activate. The gating network determines the top $n \leq M$ experts to use for a given input, and the final output is computed as a weighted sum of the selected experts' outputs:

$$Q = \sum_{i \in \mathcal{S}} g_i(Q^i)Q^i, \qquad K = \sum_{i \in \mathcal{S}} g_i(K^i)K^i, \qquad V = \sum_{i \in \mathcal{S}} g_i(V^i)V^i, \tag{7}$$

where $\mathcal{S}$ is the set of selected experts, $g_i(\cdot)$ is the gating function that assigns a weight to each expert. This combination of expert specialization and dynamic routing allows MoE models to scale efficiently by being sparse and only selecting a subset of experts to pass through.

## 3 Epistemic Mixture of Experts

Epistemic uncertainty is a cornerstone in the machine learning community for evaluating confidence in a model's predictions (Gruber et al., 2023; Wang & Ji, 2024). EMoE leverages ensembles to estimate epistemic uncertainty, following the approach of Lakshminarayanan et al. (2017). By utilizing

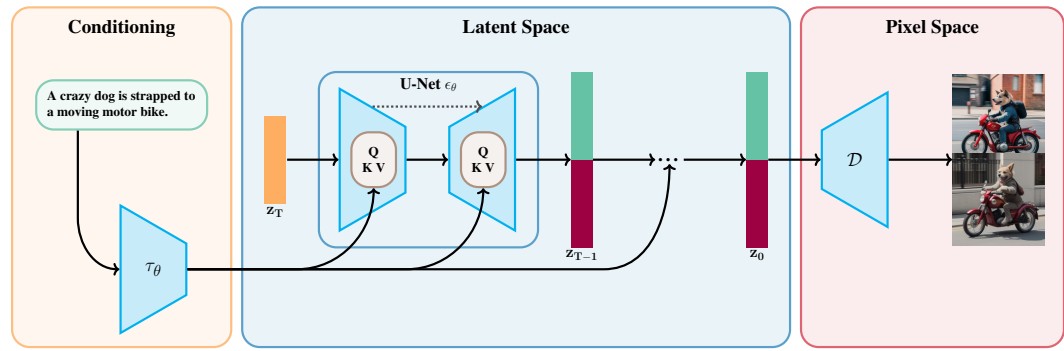

Figure 2: EMoE disentangles the expert components in the first cross-attention layer and then processes each component as a separate MoE pipeline. Thus after the first U-Net, $M$ separate latent representations are made. Illustrated is an ensemble with 2 expert components (🟩 and 🟥).

multiple models, EMoE captures the variance between model predictions, providing more reliable uncertainty estimates based on ensemble disagreement.

### 3.1 DISENTANGLING MoE

To estimate uncertainty, the ensemble components must be disentangled. In our framework, this occurs at the first mixture layer, which is the initial cross-attention layer in the first down block. Instead of aggregating the experts' outputs via a weighted sum, we create $M$ separate computational paths, each corresponding to one expert. Each path independently processes its own copy of the latent representations. Subsequent MoE layers in each branch follow the standard process, using a weighted sum of the latent representations. This process is illustrated in Figure 2 and Figure 3, where $Q^i$, $K^i$ and $V^i$ denote the different ensemble components. Note that $CA^i$ denotes the cross-attention output from the $i$th component Attention($Q^i, K^i, V^i$). This design keeps the ensemble components distinct throughout the network, enabling effective capture of diversity among the experts' predictions.

Separating the ensemble components early in the pipeline generates multiple predictions within the latent spaces of the denoising process. This enables the estimation of their disagreement (epistemic uncertainty) at the initial step of the denoising without requiring a complete forward pass through the U-Net, offering the advantage of halting the denoising process immediately for uncertain prompts. Diffusion models carry the drawback of being computationally expensive during image generation. This limitation has spurred considerable research into accelerating the denoising process (Huang et al., 2022; Wu et al., 2023). The fast computation of epistemic uncertainty in our approach aligns with ongoing efforts to reduce the environmental impact of large machine learning models (Henderson et al., 2020).

Figure 3: First cross-attention layer where EMoE disentangles the ensemble components, after which each $CA^i$ is processed as it would be in an MoE framework.

### 3.2 EPISTEMIC UNCERTAINTY ESTIMATION

After creating $M$ distinct outputs from our model, we still need to accurately capture their disagreement. For this we apply two techniques. Firstly, we capture epistemic uncertainty by measuring the variance among the ensemble components, a common approach in the literature (Ekmekci & Cetin, 2022; Chan et al., 2024).. Secondly, we estimate uncertainty after the mid-block in our U-Net, $m_0^{post}$. Note that given that this is a high-dimensional space $d_{mid}$ ($1280 \times 8 \times 8$) and we want to reduce epistemic uncertainty to one number, we take the mean across the variance of each

dimension. Thus our estimate of epistemic uncertainty is,

$$\text{EU}(y) = \mathbb{E}_{d_{mid}} \left[ \text{Var}_{i \in M} \left[ m_0^{post} \right] \right]. \tag{8}$$

It is important to note that $m_0^{post}$ takes as input the text prompt, $y$. Thus $\text{EU}(y)$ gives an estimate of the epistemic uncertainty of our MoE given a prompt $y$. The intuition behind this choice of epistemic uncertainty estimator is detailed in Appendix C.

### 3.3 BUILDING MoE

To build an ensemble that effectively captures uncertainty, the ensemble components must be diverse enough to reflect meaningful disagreement among them. In deep learning, two primary techniques have been used to achieve diversity among ensemble components: bootstrapping samples during training and random initialization (Breiman, 2001; Lakshminarayanan et al., 2017). In our approach, the ensemble components are not trained; instead, they are sourced from pre-existing models available on Hugging Face and Civit AI. This strategy offers the significant advantage of enabling the creation of large-scale ensembles, as Hugging Face hosts over 30,000 model checkpoints and Civit AI provides thousands of models.

The drawback of not controlling the training process is that ensuring sufficient diversity within the ensemble becomes largely a matter of chance. Fortunately, the wide array of models available on Hugging Face and Civit AI includes many trained for specific tasks, which naturally contributes to ensemble diversity. In contrast, training such an ensemble from scratch with these qualities would require a significant amount of computational resources.

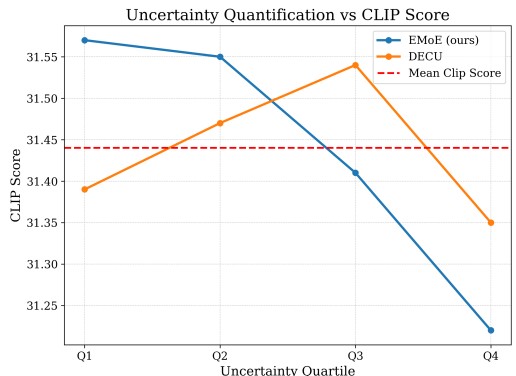

Figure 4: CLIP Score across different uncertainty quartiles. EMoE accurately attributes prompts that produce images with high CLIP scores with low uncertainty unlike Diffusion Ensembles for Capturing Uncertainty (DECU). The red line indicates the average CLIP score across all quartiles.

Table 1: Mean Length of English Prompts by Quartile of Uncertainty $\pm$ standard deviation.

| Quartile | Character Count | Word Count |
|----------|-----------------|------------|
| Q1 | $53.14 \pm 13.50$ | $10.58 \pm 2.56$ |
| Q2 | $52.38 \pm 12.94$ | $10.47 \pm 2.42$ |
| Q3 | $52.20 \pm 12.81$ | $10.43 \pm 2.39$ |
| Q4 | $51.93 \pm 12.32$ | $10.34 \pm 2.33$ |

Finally, after assembling the ensemble, a gating module is essential to route the inputs to a subset of components and weigh their outputs. While the gating module can be trained, it is also possible to infer it by using inputs that are representative of the datasets each expert was trained or fine-tuned on. As the focus of our experiments is on generative text-to-image models, these representative inputs consist of generic positive and negative input text prompts. With these inputs, we can construct *gate vectors* using the pre-trained models (e.g. using the activations of their text encoders). When a new input prompt is presented to the ensemble, the gating module compares the input activations to the gating module with the precomputed gate vectors, assigning weights to the experts based on similarity. This approach enables the construction of a MoE model that dynamically selects and weighs experts without additional training, effectively leveraging the strengths of pre-trained models to handle diverse tasks, and enabling our uncertainty estimation method. Further details can be found in Appendix D or in Goddard et al. (2024).

## 4 RESULTS

To validate EMoE, we conducted a series of experiments on the COCO dataset (Lin et al., 2014). Our codebase is built on the diffusers and segmoe libraries (von Platen et al., 2022; Yatharth Gupta, 2024), with modifications to support our method. We used the base MoE in the segmoe library, the model card for which is contained in Appendix E. For generating COCO prompts in multiple languages, we utilized the Google Translate API. Our results used the Contrastive Language-Image Pre-training (CLIP) score as a metric to evaluate how well the model aligns the generated image

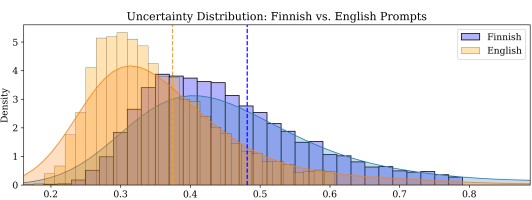

Figure 5: Uncertainty distribution for Finnish and English prompts, showing higher uncertainty for Finnish prompts compared to English.

Table 2: Comparison of CLIP scores and mean uncertainty $\pm$ standard deviation between Finnish and English prompts. Illustrating lower image quality and higher uncertainty for Finnish prompts.

| Language | CLIP Score | Uncertainty |
|---|---|---|
| Finnish | 16.41 | $0.48 \pm 0.19$ |
| English | 31.39 | $0.37 \pm 0.14$ |

with the given prompt (Hessel et al., 2021). A higher CLIP score indicates a closer semantic match between the image and the prompt. The code and dataset will be made public upon publication. Note that when evaluating the CLIP score for images generated from non-English prompts, the English version of the prompt was used for assessment. This was done to account for the fact that CLIP was primarily trained on English data.

### 4.1 ENGLISH PROMPTS

The first experiment assessed EMoE's ability to distinguish between in-distribution prompts that produce higher-quality images. We randomly sampled 40,000 prompts from the COCO dataset and calculated their epistemic uncertainty using EMoE. These prompts were then divided into four quartiles based on uncertainty: Q1, containing the lowest 25% uncertainty prompts, through Q4, representing the highest 25% uncertainty. For each bin, we generated images and evaluated their quality using

Table 3: Mean Length of Finnish Prompts by Quartile of Uncertainty.

| Quartile | Character Count | Word Count |
|---|---|---|
| Q1 | $54.94 \pm 17.04$ | $6.59 \pm 2.16$ |
| Q2 | $51.26 \pm 14.40$ | $6.14 \pm 1.79$ |
| Q3 | $49.67 \pm 14.23$ | $5.95 \pm 1.75$ |
| Q4 | $47.97 \pm 13.86$ | $5.77 \pm 1.73$ |

the CLIP score. As shown in Figure 4, there is a clear relationship between lower uncertainty (i.e., Q1) and CLIP score, while prompts in Q4 produced a lower CLIP score. These findings confirm EMoE's effectiveness in uncertainty-driven image quality estimation, demonstrating its ability to perform refined uncertainty estimation on in-distribution samples. Given that each expert has been trained on all data in the COCO dataset, EMoE's ability to detect subtle differences in uncertainty on in-sample data is a notable feature. In contrast, the DECU baseline (Berry et al., 2024) did not demonstrate this capability.

We further analyzed prompt characteristics across uncertainty quartiles. Prompts in the lower uncertainty quartiles (i.e., Q1 and Q2) were shorter in both character and word count, as shown in Table 1. This aligns with the intuition that longer prompts are more descriptive, providing the model with clearer objectives. These results further underscore EMoE's ability to capture uncertainty as expected, highlighting its robustness in managing in-distribution prompt variations.

### 4.2 FINNISH PROMPTS

Next, to assess EMoE's ability to differentiate between in-distribution and out-of-distribution samples, we translated 10,000 English prompts to Finnish. Given Finnish's lower representation in online datasets, we expected Finnish prompts to be more likely out-of-distribution, resulting in lower image quality. As shown in Figure 5, the uncertainty distribution for Finnish prompts is skewed more to the right than for English prompts, demonstrating EMoE's capability to distinguish between in- and out-of-distribution samples. The relationship between CLIP score and uncertainty is detailed in Table 2. In line with Table 1, we observed that longer prompts are associated

Table 4: Comparison of the proportion of prompts with "pizza" in Q1 of uncertainty between Finnish and English prompts.

| Language | Proportion of Prompts with "pizza" in Q1 |
|---|---|
| Finnish | 46.67% |
| English | 21.54% |

with lower uncertainty, even for out-of-distribution samples, as shown in Table 3. This suggests that even in unfamiliar languages, longer prompts give the model more confidence in its output.

**Neliön muotoinen pizza, jonka henkilö leikkaa isolla veitsellä**

**A square shaped pizza being cut by a person with a big knife.**

**Hääkakku on valkoinen ja siinä on kukkia.**

**The wedding cake is white with flowers on it.**

Figure 6: Qualitative comparison of image-generation for a Finnish prompt with the word "pizza" and a random Finnish prompt. Note that the English translation was not provided to the model.

We also leveraged EMoE to detect bias within the model. During our analysis of images generated from Finnish prompts, prompts containing the word "pizza" consistently produced more text-aligned images as opposed to random prompts, as illustrated in Figure 6. Results from EMoE also supported this relationship, with 46.67% of Finnish "pizza" prompts falling into the lowest uncertainty quartile (Q1), compared to only 21.54% for English prompts, as seen in Table 4.

### 4.3    MULTI-LINGUAL PROMPTS

To further explore the behavior of EMoE, we translated 1,000 prompts into an additional 23 languages via Google Translate. We applied EMoE to these translations and calculated each language's respective CLIP score. As shown in Figure 7, there is a strong negative correlation ($r = -0.79$) between uncertainty (as measured by EMoE) and CLIP score, consistent with the expected relationship between uncertainty and image quality. Additionally, the size of each point in Figure 7 is proportional to the number of native speakers for each language. One can also observe a relationship between the number of native speakers with both CLIP score and uncertainty of any given language. European

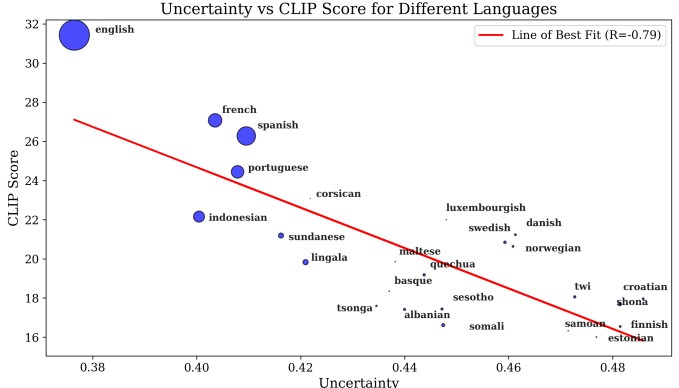

Figure 7: Negative correlation between uncertainty and image quality across prompts translated into 25 different languages. EMoE demonstrates a strong negative correlation (r = -0.79) between uncertainty and CLIP score, with languages having more native speakers generally producing lower uncertainty and higher-quality images, highlighting potential biases in text-to-image models favoring more commonly spoken languages.

languages generally performed better than non-European languages, which further underscores the potential bias in favor of European languages in text-to-image models and EMoE's ability to capture language related model bias. This section and Section 4.2 illustrate the model's bias toward certain languages and reveal its unfairness toward non-European languages. This demonstrates how EMoE can be utilized to detect biases and identify the data necessary for training to mitigate these issues.

### 4.4    ABLATION

We conducted 4 ablation experiments to determine the optimal number of ensemble components, the effect of the denoising step for estimating uncertainty, the most suitable latent space for uncertainty estimation, and we evaluated EMoE on another MoE model to validate the robustness of our approach. All ablation studies were performed on a dataset of 40,000 English prompts.

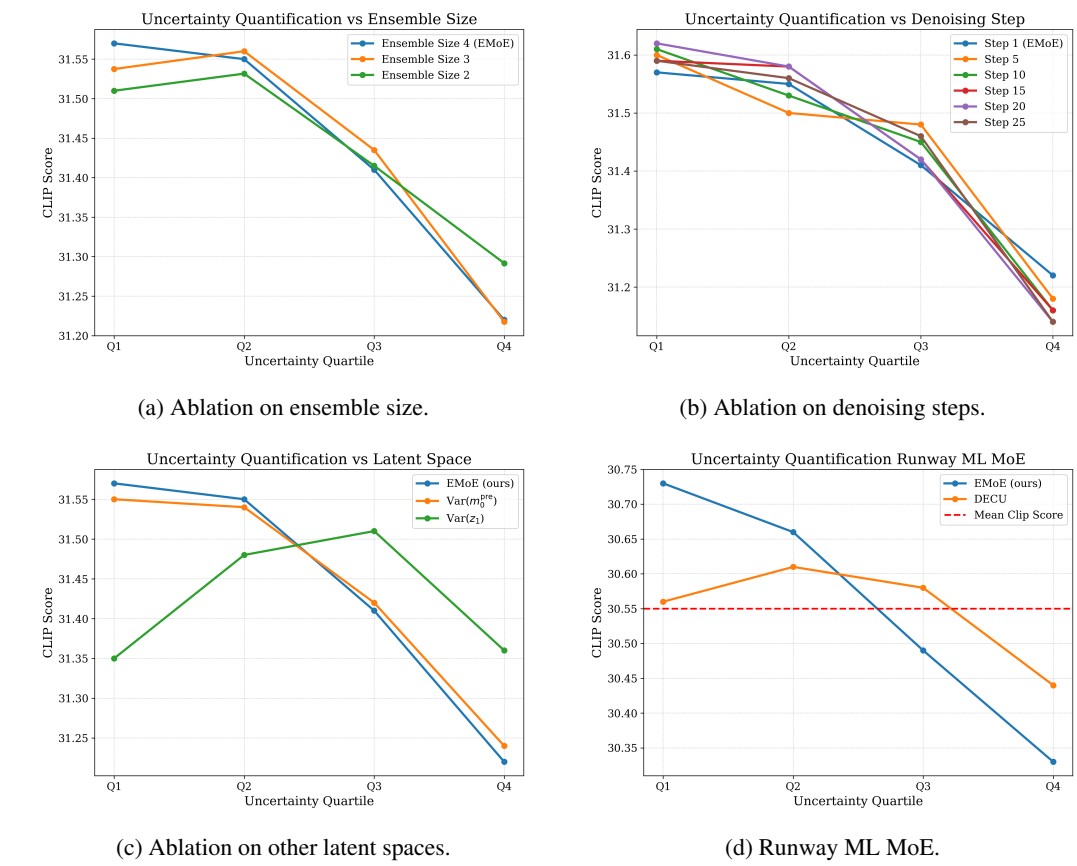

Figure 8: Ablation studies validating EMoE hyperparameters: ensemble size (a), denoising step (b), and latent space (c). Additionally, (d) shows the robustness of EMoE using Runway MoE.

To identify the optimal number of ensemble components, we examined ensemble sizes of 2 and 3, using all possible permutations from the 4 components. We averaged the results for ensembles of 2 and 3 components (Figure 8a). The results indicate that ensemble sizes of 2 and 3 are sub-optimal to an ensemble size of 4, as the first quantile (Q1) yields a lower CLIP score than the second (Q2).

We investigated the effect of the denoising step on uncertainty quantification, as shown in Figure 8b. A consistent decrease in CLIP scores across uncertainty quantiles at each step confirmed EMoE's robustness in estimating epistemic uncertainty. For practical reasons, we selected the first step, as it offers the earliest opportunity to halt the costly denoising process for high-uncertainty prompts.

We also explored different latent spaces in which to estimate epistemic uncertainty, testing both $Var(m_0^{pre})$ and $Var(z_1)$. The results, shown in Figure 8c, indicate that $Var(z_1)$ is sub-optimal, aligning with previous findings from DECU. We observed that $Var(m_0^{pre})$ performed similarly to $Var(m_0^{post})$. We chose $Var(m_0^{post})$ because the mid-block is intended to refine the latent space, though $Var(m_0^{pre})$ could serve as an acceptable alternative.

Finally, to further validate the robustness of EMoE, we ran an additional experiment using Runway MoE (Figure 8d). The results confirm that EMoE is versatile and can effectively handle different MoE models. Additionally, this demonstrates that EMoE can detect uncertainty even within the context of very similar models as each expert component is a version of Runway ML stable diffusion.

## 5 RELATED WORKS

Building ensembles of diffusion models for advanced image generation is challenging due to the large number of parameters, often exceeding hundreds of millions (Saharia et al., 2022; Nichol et al., 2022; Ramesh et al., 2022). Despite this, methods like eDiff-I have emerged, using ensem-

ble techniques to enhance image fidelity, though not for epistemic uncertainty estimation, requiring approximately 2 million training iterations (Balaji et al., 2022). In contrast, DECU was specifically developed for uncertainty estimation, with a training duration of 7 days (Berry et al., 2024), and focuses on estimating epistemic uncertainty for class label image generation. Our approach, however, leverages pre-trained experts for epistemic uncertainty estimation, thereby reducing the computational burden to zero. Moreover, EMoE addresses a more complex challenge—estimating epistemic uncertainty in text-based generation, rather than in a discrete input like a class label.

Previous research has addressed epistemic uncertainty estimation in neural networks, particularly for image classification tasks, by employing Bayesian approximations (Gal et al., 2017; Kendall & Gal, 2017; Kirsch et al., 2019). These works focus on discrete output spaces, which are significantly simpler than image generation. However, another approach to estimating epistemic uncertainty is the use of ensembles (Lakshminarayanan et al., 2017; Choi et al., 2018; Chua et al., 2018), commonly applied in regression tasks (Depeweg et al., 2018; Postels et al., 2020; Berry & Meger, 2023a;b). For example, Postels et al. (2020) and Berry & Meger (2023b) developed efficient ensemble generative models based on Normalizing Flows (NF) to capture epistemic uncertainty. Berry & Meger (2023a) further advanced these methods by using Pairwise Difference Estimators to estimate uncertainty in a 257-dimensional output space with normalizing flows. Our work builds on this foundation by extending these techniques to higher-dimensional outputs (786,432 dimensions) in large diffusion models and considering the more complex input space of text.

With the rise of large generative models and the growing importance of uncertainty estimation, numerous methods have been developed to estimate uncertainty in both image and text generation models (Malinin & Gales, 2020; Berry et al., 2024; Chan et al., 2024; Liu et al., 2024). For instance, Chan et al. (2024) trained hyper-networks to estimate uncertainty in diffusion models for weather prediction. In contrast, EMoE generates uncertainty estimates from pre-trained expert networks, which are widely available online, such as on platforms like Hugging Face and Civit AI. Additionally, some researchers have proposed using epistemic uncertainty to detect hallucinated responses from large language models (Verma et al., 2023). In this context, EMoE could be employed for hallucination detection in vision-language models, although the definition of hallucinated responses varies across the literature (Xu et al., 2024; Duan et al., 2024; Sky et al., 2024). Further, while previous methods have integrated uncertainty into model pipelines using MoE (Zheng et al., 2019; Luttner, 2023; Zhang et al., 2024), these approaches neither address epistemic uncertainty nor consider text-to-image generation tasks and are not applicable in a zero-shot manner.

## 6 CONCLUSIONS

In this paper, we introduced the Epistemic Mixture of Experts (EMoE) framework for estimating uncertainty in text-to-image diffusion models. EMoE leverages pre-trained experts to provide computationally efficient uncertainty estimates without the need for additional training. By incorporating a novel latent space for uncertainty estimation within the diffusion process, EMoE can identify biases and regions of heightened uncertainty early in the image generation process.

**Limitations**. EMoE relies on the availability of pre-trained expert networks, which, although abundant, may not always provide sufficient diversity for optimal uncertainty estimation in all scenarios. The framework's performance is closely linked to the quality and diversity of the pre-trained models it uses, which introduces potential unpredictability when handling novel or specialized inputs. Furthermore, while EMoE does not require additional training, it does require sufficient memory resources to load and run the ensemble of experts effectively.

Our experimental results show that EMoE not only improves the detection of epistemic uncertainty but also sheds light on underrepresented linguistic biases in diffusion models. By utilizing readily available pre-trained models, we demonstrated that EMoE scales efficiently while delivering reliable uncertainty estimates across a variety of input prompts. These capabilities have significant implications for fairness, accountability, and the robustness of AI-generated content.

As large generative models continue to expand in use, the ability to quantify and interpret uncertainty will be increasingly important, particularly in applications like autonomous systems. Future work may explore ways to address the limitations discussed and further optimize EMoE for more complex tasks and environments.

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

## A    COMPUTE DETAILS

We used the same set of hyperparameters as in the Stable Diffusion model described by Yatharth Gupta (2024). Minor changes were made to both the Segmoe and Diffusers codebases to disentangle the MoE, with specific modifications to incorporate EMoE. Our infrastructure included an AMD Milan 7413 CPU running at 2.65 GHz, with a 128M L3 cache, and an NVIDIA A100 GPU with 40 GB of memory. The wall clock time required to collect each dataset and the memory usage are provided in Figure 9. The parameter count for the Segmoe model is 1.63 billion parameters, while a single model contains 1.07 billion parameters. This highlights the efficiency of using a sparse MoE approach compared to creating 4 distinct models, as the Segmoe model is only 153% the size of a single model, rather than 400%. When running the SegMoE model in its standard mode, generating an image from one prompt takes an average of 3.58 seconds. In comparison, using EMoE typically requires an average of 12.32 seconds to generate four images from a single prompt. However, for scenarios where only one image per prompt is needed, EMoE's output can be optimized by estimating epistemic uncertainty during the initial diffusion step, followed by standard MoE-based image generation. This optimized version of EMoE, Fast EMoE, achieves an average generation time of 5.5 seconds. Figure 10 provides further details. Note that uncertainty reported across all experiments is calculated as $\sqrt{d_{midsize}} \times \text{EU}(y)$, where $d_{midsize} = 1280 \times 8 \times 8$.

Figure 9: Computational requirements.

| Dataset | Run Time | Storage |
|---|---|---|
| English 40k Prompts | 200 gpu hrs | 6 TB |
| Finnish 10k Prompts | 50 gpu hrs | 1.5 TB |
| Other Languages 1k Prompts | 5 gpu hrs | 150 GB |

Figure 10: Generation times for baseline (Segmoe) and two variants of EMoE. Reported times are $\mu \pm \sigma$.

| Model | Generation Time |
|---|---|
| Segmoe | $3.58 \pm 0.54$ secs |
| EMoE | $12.32 \pm 4.6$ secs |
| Fast EMoE | $5.5 \pm 0.15$ secs |

---

**Algorithm 1** Epistemic Mixture of Experts (EMoE)

1: **Input:** Initial noise $z_T \sim \mathcal{N}(\mathbf{0}, \mathbf{I})$, total steps $T$, pre-trained experts $E = \{e_1, e_2, \ldots, e_M\}$, prompt $y$
2: **for** $t = T$ to $1$ **do**
3:    **if** $t = T$ **then**
4:        **Disentangle Experts:**
5:        **for** each expert $e_i \in E$ **do**
6:            Pass $z_T$ and prompt $y$ through $e_i$'s first cross-attention layer to arrive at $M$ distinct generations (Figure 3).
7:            Extract the mid-block latent representation $m_0^{post,i}$.
8:        **end for**
9:        Compute epistemic uncertainty $\text{EU}(y)$ as defined in Equation 8.
10:        Output $M$ different $\mathbf{z}_{t-1}^i$, one for each expert.
11:    **else**
12:        **Mixture of Experts Rollout:**
13:        **for** $i \in \{1, ..., M\}$ **do**
14:            Update latent variable for each expert:

$$\mathbf{z}_{t-1}^i \sim p(\mathbf{z}_{t-1}^i | \mathbf{z}_t^i, y)$$

15:            Pass $\mathbf{z}_t^i$ and $y$ through our MoE without disentangling, as shown in Figure 2 in ▮ and ▮.
16:        **end for**
17:    **end if**
18: **end for**
19: **Output:** $M$ reconstructed latent variables $\mathbf{z}_0^i$ and $\text{EU}(y)$.

## B    BIAS IN CLIP SCORE

The CLIP score, despite its known biases (Chinchure et al., 2023), remains a widely-used method for evaluating the alignment between text prompts and generated images, alongside FID (Shi et al., 2020; Kumari et al., 2023). Both metrics, however, rely on auxiliary models (CLIP and Inception, respectively), making them susceptible to inherent biases. While FID requires a large number of samples for reliable estimation, the CLIP score facilitates a more direct assessment of text-to-image alignment with fewer samples (Kawar et al., 2023; Ho et al., 2022a). Considering these trade-offs, we prioritized the CLIP score due to its relevance to our research objectives and its broad acceptance in related studies.

Table 5: SSIM on each uncertainty quartile, using EMoE, in the English 40k dataset.

| Quartile | SSIM |
|----------|-------|
| Q1 | 0.234 |
| Q2 | 0.231 |
| Q3 | 0.228 |
| Q4 | 0.226 |

To further validate our findings and address any potential concerns related to metric biases, we conducted an additional experiment using the Structural Similarity Index (SSIM) as the evaluation metric. Unlike CLIP or FID, SSIM does not depend on any auxiliary models for its calculation, thereby mitigating the risk of bias. We computed SSIM between generated images and corresponding ground-truth images from the COCO dataset and analyzed the results for each uncertainty quartile. As shown in Table 5, EMoE effectively categorized prompts into the appropriate uncertainty quartiles based on model performance. This provides further evidence of EMoE's efficacy in estimating uncertainty for MoE text-to-image models, highlighting its robustness across different evaluation metrics.

## C    INTUITION BEHIND OUR ESTIMATOR FOR EPISTEMIC UNCERTAINTY

Here is an intuitive explanation for our choice of estimator for epistemic uncertainty using the theory of Gaussian Processes. Each expert can be viewed as a sample from the posterior distribution of functions given an input $y$, denoted as $p(f(y)|y)$. By calculating the variance across these experts, we obtain the variance $\sigma^2$ of $p(f(y)|y)$, which serves as an estimate of epistemic uncertainty within the Gaussian Process framework. In general, other works have used the difference among ensemble components to denote epistemic uncertainty (Gal et al., 2017; Depeweg et al., 2018; Berry & Meger, 2023b).

When estimating the epistemic uncertainty for a prompt $y$, we weight each ensemble component equally. Therefore, let $\mathcal{F} = \{f_{\theta_i}\}_{i=1}^N$ denote an ensemble of $N$ neural networks, where each model $f_{\theta_i} : \mathcal{Y} \to \mathbb{R}$ is parameterized by $\theta_i$, sampled from a parameter distribution $p(\theta)$. Then the prediction from our ensemble is:

$$\hat{f}(y) = \frac{1}{N} \sum_{i=1}^N f_{\theta_i}(y),$$

where $y \in \mathcal{Y}$ is an input from the input space $\mathcal{Y}$.

A **Gaussian Process** (GP) is defined as a collection of random variables, any finite subset of which follows a joint Gaussian distribution. Formally, a Gaussian Process $f(y) \sim \mathcal{GP}(\mu(y), k(y, y'))$ is characterized by its mean function $\mu(y)$ and covariance function $k(y, y')$:

$$\mu(y) = \mathbb{E}[f(y)], \quad k(y, y') = \mathbb{E}[(f(y) - \mu(y))(f(y') - \mu(y'))].$$

**Proposition 1:** Let $\mathcal{F} = \{f_{\theta_i}\}_{i=1}^N$ be an ensemble of neural networks with parameter samples $\theta_i \sim p(\theta)$. As $N \to \infty$ and under the assumption that the neural network weights are drawn i.i.d. from a distribution with zero mean and finite variance, the ensemble predictor $\hat{f}(y)$ converges in distribution to a Gaussian Process:

$$\hat{f}(y) \xrightarrow{d} \mathcal{GP}(\mu(y), k(y, y')),$$

where $\mu(y)$ is the expected value of the ensemble output, and $k(y, y')$ is the covariance function defined by the variance of the ensemble.

**Proof:**

To prove this, we proceed in two main steps:

STEP 1: CONVERGENCE OF MEAN FUNCTION

Consider the mean function $\mu(y)$ of the ensemble predictor:

$$\mu(y) = \mathbb{E}_{\theta \sim p(\theta)}[f_\theta(y)].$$

As $N \to \infty$, by the law of large numbers, the empirical mean of the ensemble $\hat{f}(y)$ converges to the expected mean:

$$\lim_{N \to \infty} \frac{1}{N} \sum_{i=1}^{N} f_{\theta_i}(y) = \mu(y).$$

STEP 2: CONVERGENCE OF COVARIANCE FUNCTION

The covariance function $k(y, y')$ of the Gaussian Process can be defined as:

$$k(y, y') = \lim_{N \to \infty} \frac{1}{N} \sum_{i=1}^{N} \left(f_{\theta_i}(y) - \mu(y)\right) \left(f_{\theta_i}(y') - \mu(y')\right).$$

Under the assumption that $f_{\theta_i}(y)$ are i.i.d. samples with finite variance, by the Central Limit Theorem (CLT), the ensemble prediction $\hat{f}(y)$ converges in distribution to a Gaussian Process $\mathcal{GP}(\mu(y), k(y, y'))$.

In the context of an ensemble of neural networks, **epistemic uncertainty** arises from the uncertainty over the model parameters $\theta$. This uncertainty is captured by the variance of the ensemble predictions:

$$\mathrm{Var}[\hat{f}(y)] = \frac{1}{N} \sum_{i=1}^{N} (f_{\theta_i}(y) - \hat{f}(y))^2.$$

As $N \to \infty$, this variance converges to the posterior variance of the Gaussian Process:

$$\lim_{N \to \infty} \mathrm{Var}[\hat{f}(y)] = k(y, y),$$

where $k(y, y)$ is the marginal variance of the Gaussian Process and directly represents the **epistemic uncertainty**.

## D  GATES WITHOUT TRAINING

Each expert is associated with a positive and a negative descriptor, $Des^i = (Pos^i, Neg^i)$, which represent what the expert excels at and struggles with modeling, respectively. These descriptors are processed through a pre-trained text model, $\rho_\phi$, to create *gate vectors*, $g^i$. When a new positive and negative prompt, $y^j = (pos^j, neg^j)$, is provided to generate an image, these prompts are compared against $g^i$ and assigned a weight, $w_i$ based on the dot product. This process is illustrated in Figure 11 and described in Goddard et al. (2024).

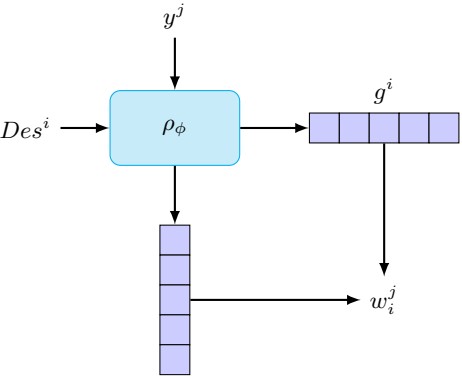

Figure 11: This pictures depicts how to have accurate gates without training.

## E  MODEL CARDS

Below are the model parameters for the base Segmoe MoE used in the experiments. We increased the number of experts from 2 to 4 to incorporate more ensemble components. Generally, having a low

number of ensemble components (2-10) is sufficient in deep learning to capture model disagreement (Osband et al., 2016; Chua et al., 2018; Fujimoto et al., 2018). In addition to the Segmoe base MoE, we also tested EMoE on another MoE model, referred to as Runway ML, where each expert component is a Runway model. The corresponding model card can be found below. This experiment demonstrates the robustness of EMoE across different MoE architectures, showing that EMoE is effective even when components are trained on similar data with similarly initialized weights, as each Runway ML component was fine-tuned on new data from similar initial conditions.

```
Segmoe MoE

base_model: SG161222/Realistic_Vision_V6.0_B1_noVAE
num_experts: 4
moe_layers: all
num_experts_per_tok: 2
type: sd
experts:
  - source_model: SG161222/Realistic_Vision_V6.0_B1_noVAE
    positive_prompt: "cinematic, portrait, photograph, instagram,
        fashion, movie, macro shot, 8K, RAW, hyperrealistic, ultra
         realistic,"
    negative_prompt: " (deformed iris, deformed pupils, semi-
        realistic, cgi, 3d, render, sketch, cartoon, drawing,
        anime), text, cropped, out of frame, worst quality, low
        quality, jpeg artifacts, ugly, duplicate, morbid,
        mutilated, extra fingers, mutated hands, poorly drawn
        hands, poorly drawn face, mutation, deformed, blurry,
        dehydrated, bad anatomy, bad proportions, extra limbs,
        cloned face, disfigured, gross proportions, malformed
        limbs, missing arms, missing legs, extra arms, extra legs,
         fused fingers, too many fingers, long neck"
  - source_model: dreamlike-art/dreamlike-anime-1.0
    positive_prompt: "photo anime, masterpiece, high quality,
        absurdres, 1girl, 1boy, waifu, chibi"
    negative_prompt: "simple background, duplicate, retro style,
        low quality, lowest quality, 1980s, 1990s, 2000s, 2005
        2006 2007 2008 2009 2010 2011 2012 2013, bad anatomy, bad
        proportions, extra digits, lowres, username, artist name,
        error, duplicate, watermark, signature, text, extra digit,
         fewer digits, worst quality, jpeg artifacts, blurry"
  - source_model: Lykon/dreamshaper-8
    positive_prompt: "bokeh, intricate, elegant, sharp focus, soft
         lighting, vibrant colors, dreamlike, fantasy, artstation,
         concept art"
    negative_prompt: "low quality, lowres, jpeg artifacts,
        signature, bad anatomy, extra legs, extra arms, extra
        fingers, poorly drawn hands, poorly drawn feet, disfigured
        , out of frame, tiling, bad art, deformed, mutated, blurry
        , fuzzy, misshaped, mutant, gross, disgusting, ugly,
        watermark, watermarks"
  - source_model: dreamlike-art/dreamlike-diffusion-1.0
    positive_prompt: "dreamlikeart, a grungy woman with rainbow
        hair, travelling between dimensions, dynamic pose, happy,
        soft eyes and narrow chin, extreme bokeh, dainty figure,
        long hair straight down, torn kawaii shirt and baggy jeans
        , In style of by Jordan Grimmer and greg rutkowski, crisp
        lines and color, complex background, particles, lines,
        wind, concept art, sharp focus, vivid colors"
    negative_prompt: "nude, naked, low quality, lowres, jpeg
        artifacts, signature, bad anatomy, extra legs, extra arms,
         extra fingers, poorly drawn hands, poorly drawn feet,
        disfigured, out of frame"
```

```
Runway ML MoE

base_model: runwayml/stable-diffusion-v1-5
num_experts: 4
moe_layers: all
num_experts_per_tok: 4
type: sd
experts:
  - source_model: runwayml/stable-diffusion-v1-5
    positive_prompt: "ultra realistic, photos, cartoon characters,
        high quality, anime"
    negative_prompt: "faces, limbs, facial features, in frame,
        worst quality, hands, drawings, proportions"
  - source_model: CompVis/stable-diffusion-v1-4
    positive_prompt: "ultra realistic, photos, cartoon characters,
        high quality, anime"
    negative_prompt: "faces, limbs, facial features, in frame,
        worst quality, hands, drawings, proportions"
  - source_model: CompVis/stable-diffusion-v1-3
    positive_prompt: "ultra realistic, photos, cartoon characters,
        high quality, anime"
    negative_prompt: "faces, limbs, facial features, in frame,
        worst quality, hands, drawings, proportions"
  - source_model: CompVis/stable-diffusion-v1-2
    positive_prompt: "ultra realistic, photos, cartoon characters,
        high quality, anime"
    negative_prompt: "faces, limbs, facial features, in frame,
        worst quality, hands, drawings, proportions"
```

## F  QUALITATIVE RESULTS

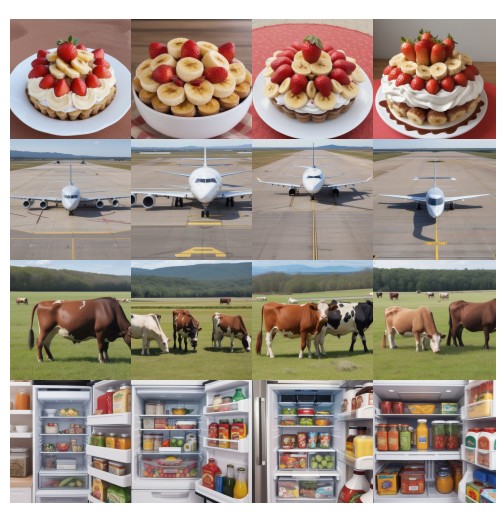 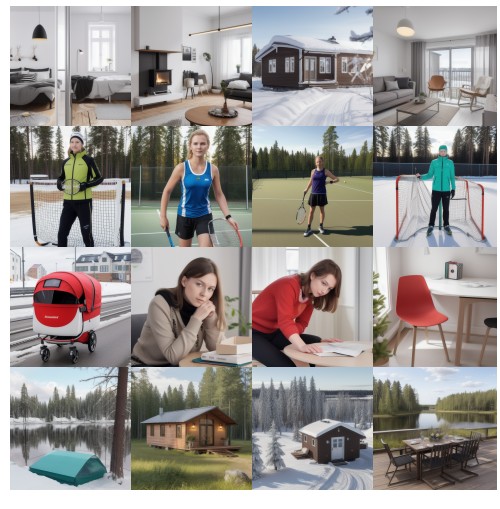

**Low Epistemic Uncertainty**          **High Epistemic Uncertainty**

Figure 12: EMoE's uncertainty across different prompts: Each row represents a distinct prompt, while the columns denote the output of each component. The left panel displays low uncertainty, while the right panel shows higher uncertainty, indicating more ambiguous or less familiar prompts.

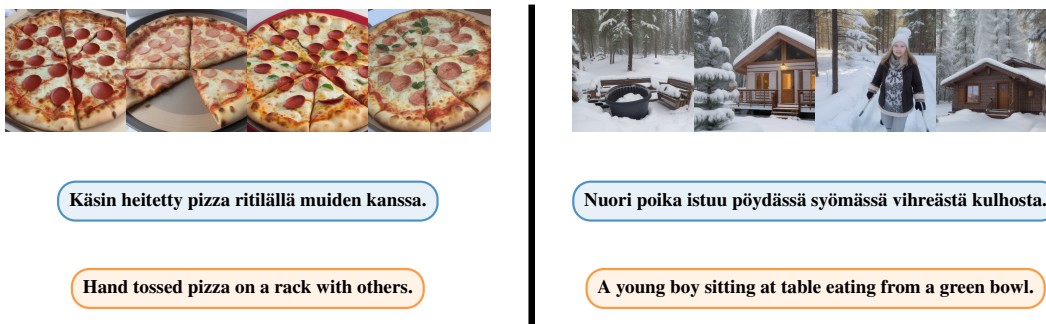

Figure 13: Qualitative comparison of image-generation for a Finnish prompt with the word "pizza" and a random Finnish prompt. Note that the English translation was not provided to the model.

In addition to the examples provided in the main paper, we have included additional qualitative results of our MoE model. Figure 12 shows two sets of images: low uncertainty images on the left and high uncertainty images on the right. Each row corresponds to a single prompt, while the columns display the outputs from different ensemble components. The low uncertainty prompts exhibit less variation across ensemble outputs, whereas the high uncertainty prompts show greater diversity, indicating the model's difficulty in capturing the semantic meaning of the prompt in the generated images. Here, we present another example of models showing bias towards Finnish prompts containing "pizza", as illustrated in Figure 13.

