# OpenReview forum: "Seeing the Unseen: How EMoE Unveils Bias in Text-to-Image Diffusion Models"
_ICLR.cc/2025/Conference — Submitted to ICLR 2025_

### Official Review · Reviewer_MQCg · 2024-10-28

**Soundness:** 2
**Presentation:** 1
**Contribution:** 2
**Rating:** 3
**Confidence:** 2

**Summary:**

The paper proposes EMOE, a method to measure epistemic uncertainty (EU) in text-to-image diffusion models using pre-trained components. Experiments show that higher EU relates to poorer image quality and inconsistent outputs. The authors also carry out experiments to measure societal biases such as language and gender using EU.

**Strengths:**

- Using pretrained components to measure EU is more resource-efficient than training the MOE in baselines.
- Showing that EU can be used to reveal biases such as on language and gender is highly relevant given the increasing interest in safe and fair generative models.

**Weaknesses:**

- My main issue with the paper is the explanation of how the method works, which could stem from my unfamiliarity with MoE models. Are the authors using M different diffusion models and decoding the same prompt, then measuring the variance between models at a predefined layer of the Unet block to calculate EU? Fig 2 and 3 are not clear to me.

- As I understand, MoE requires training of a gating layer and the experts, while the authors appear to use a pretrained library (segmoe) that can accomodate any combination of different diffusion models without explanation of how it works. Could the authors clarify how segmoe works? I suggest including an explanation in the paper as well.

- In paragraph lines 306-315, how does the gating mechanism work? The method 'compares the input activations to the gating module with the precomputed gate vectors", but theres no further details of what this means. Can the authors either provide pseudocode (or concrete mathematical expressions if they differ from vanilla MOE) of the gating mechanism? Are these modifications what the authors are referring to in line 321: "with modifications to support our method"?

- I believe providing pseudocode of the modifications they made and the inference process to measure EU will improve clarity of the paper substantially.

**Questions:**

- What does CA refer to in Figure 3 caption?
- Typo on line 353-354: "Prompts in the lower uncertainty quartiles (i.e., Q1 and Q2) were shorter in both character and word count" but it is the other way around in Table 1.
- Is it normal for the CLIP score variation to be so small and are such small variations meaningful? E.g. in Fig 4, the left axis scale is so small (0.3 between the min and max points) while its very large in Fig 7.
- I'm not sure what Fig 6 is showing and what the English prompts are for. Without any basis for comparison like qualitative samples of lower quality images or quantitative metrics we cannot conclude whether those images are 'high' or 'low' quality in line 376.

---

> ### Author Response · Authors · 2024-11-19
>
> ## **Dear Reviewer,**
>
> Thank you for your insightful comments. Please find below our responses and the corresponding changes, which have also been highlighted in the provided PDF document in red.
>
> ### **Comment regarding MoE unfamiliarity and Pseudocode**
>
> MoE creates an efficient ensemble within a submodule of the generation pipeline by taking predictions from each expert—viewed as individual ensemble components—and aggregating them into a final output. EMoE separates this aggregation process, treating the outputs from each expert independently to estimate epistemic uncertainty. Rather than creating $M$ distinct models, we leverage the experts as our ensemble within a network submodule, making our approach significantly more efficient in terms of both computation and parameter count. To clarify this further, we have added the following text to the paper:
>
> >The parameter count for the Segmoe model is 1.63 billion parameters, while a single model contains 1.07 billion parameters. This highlights the efficiency of using a sparse MoE approach compared to creating 4 distinct models, as the Segmoe model is only 153% the size of a single model, rather than 400%.
>
> EMoE then evaluates the discrepancies among these experts at a predefined stage in the reverse diffusion process. For clarity, we have also added pseudocode in Appendix, Algorithm 1 with reference to Figure 2 and 3.
>
> ### **Comment regarding gating**
>
> The gating network used in our mixture of experts approach is not novel to this paper and follows established practices within the community. In our setup, each expert is associated with a positive and a negative descriptor prompt. The positive descriptor highlights the expert’s strengths, while the negative descriptor outlines its limitations. When a positive and negative prompt are provided for image generation, the system calculates gating weights for each expert based on the distance between the input prompts and the corresponding descriptor prompts in a latent space weighting them. To provide further clarity, we have included a detailed explanation in the appendix.
>
> ### **Comments regarding typos**
>
> Thank you for pointing these out, we have changed shorter to longer and defined $CA^i$ in the paper:
>
> >Note that $CA^i$ denotes the cross-attention output from the $i$th component Attention($Q^i, K^i, V^i$).
>
> ### **Comment regarding small deviation in CLIP score**
>
> The relatively small differences in CLIP scores in Figure 4 and Table 1 can be attributed to each expert's prior exposure to the training data from Q1, Q2, Q3, and Q4. Consequently, well-trained experts are expected to consistently generate high-quality images for all prompts within the training distribution. A notable feature of EMoE is its ability to detect subtle variations in uncertainty, even with in-sample data. To further clarify, we have added the following text:
>
> >Given that each expert has been trained on all data in the COCO dataset, EMoE’s ability to detect subtle differences in uncertainty on in-sample data is a notable feature
>
> Figure 7 underscores the languages these models may or may not have been trained on, highlighting EMoE’s capability to accurately capture model biases toward specific languages. To enhance this point, we have added the following text for clarity:
>
> >This section and Section 4.2 illustrate the model’s bias toward certain languages and reveal its unfairness toward non-European languages. This demonstrates how EMoE can be utilized to detect biases and identify the data necessary for training to mitigate these issues.
>
> ### **Comments regarding Figure 6**
>
> The purpose of this image is to demonstrate that Finnish prompts containing the word "pizza" were more text-aligned, indicating a bias in the model towards certain prompts. We acknowledge the reviewer's suggestion that the image could be clearer in conveying this point. To address this, we have updated both the image and the accompanying text for clarity. The revised image now compares a prompt featuring the word ``pizza” and a random Finnish prompt. Please note that the model received the Finnish prompt directly, while the English translation is provided for reader reference.
>
> ### **Comments regarding how Segmoe works**
>
> Given the complexity of the code, which builds on numerous submodules such as VAEs, UNets, tokenizers, attention mechanisms, diffusion processes, and more, it is not feasible to cover the entire pipeline exhaustively. Instead, we have selectively highlighted sections of the code that are crucial to explaining the distinct aspects of EMoE. We believe that the clarifying text, particularly in the appendix, will enhance the overall understanding and clarity of the code. In addition, the code is provided for full transparency.

---

> ### Comment · Reviewer_MQCg · 2024-11-23
> **Thanks for the response**
>
> I thank the authors for their response.
>
> ### On MoE/Gating explanation
> I still find the entire setup of EMOE poorly explained. To my knowledge, MoE using pretrained DMs with SegMOE is not common in the research literature, so it is important to explain clearly how the gating and MoE mechanism works. The pseudocode is a good start. I suggest the authors also include a pseudocode on the gating process, e.g., explicitly how to calculate the weights assigned to each expert.
>
> It took me some effort to realize the various experts are full diffusion models trained on different datasets, and the 'positive' and 'negative' prompts that define the gates are user-injected knowledge based on the training dataset. This is buried in App E on model cards. I think this must be more clearly stated in the main text as this definition of MoE departs significantly from the MoE referred to in the literature on LLMs, for example, where the gates are trained.
>
> How does the sparse MoE work? This seems like a crucial detail missing which forms a huge advantage of the work. I'm not sure how one can combine four diffusion models and only require 153% parameters of a single model without training some specific routing mechanism. It looks like the magic comes from the implementation of SegMOE, which is an open-source library on GitHub. However, this is a research paper, so I think crucial details like this should be explained without expecting the reader to consult code of a large library *as I believe sparse MOEs using pretrained DMs is not common in the research literature*.
>
> ### Typo
> I think the typo is still there in line 359: it says Q1 and Q2 have shorter character count.
>
> ### On Motivation
> On reading the comments by the other reviewer, I encourage the authors to more clearly motivate why measuring EU is important in text-to-image modeling beyond measuring bias in datasets, which I think is a small point. For instance, authors have hinted at this as higher EU means lower CLIP score in Fig 4. Authors can be clear about this as an advantage of measuring EU, and perhaps supplement with additional experiments showing relationship of image quality (eg. FID) and EU. Otherwise, it seems from Fig 1 that qualitatively, the model has no issues producing high-quality images that correspond to the prompt even for high EU prompts.

---

> > ### Author Response · Authors · 2024-12-04
> >
> > Dear Reviewer,
> >
> > Thank you for your feedback.
> >
> > ## MoE/Gating explanation:
> >
> > We would like to reiterate that neither the gating mechanism nor sparse Mixture of Experts (MoE) is a novel contribution of our paper. As the pipelines for these types of models continue to grow in complexity, with an increasing number of components and methods, it is not feasible to provide detailed explanations of each part of the network. To address this, we have provided references to the original source materials for readers to gain a deeper understanding of specific components, as well as the full codebase for transparency.
> >
> > It is also important to clarify that each expert in our model is not a distinct diffusion model. This design choice allows us to use only 153% of the parameters, instead of 400%, as would be required with independent diffusion models. This optimization is explained in the appendix and explicitly highlighted in red.
> >
> > Furthermore, sparse MoE architectures are now a critical component of large-scale generative models (e.g., ChatGPT). Given the widespread adoption of these methods, we assumed that researchers in the field would already be familiar with their functionality, which is why we did not delve into their specifics in great detail.
> >
> > ## Motivation:
> >
> > You are correct that epistemic uncertainty has a wide range of applications in the literature. However, we chose to highlight bias in these models because bias and unfairness can have serious consequences for underrepresented populations. We believe addressing bias is a crucial issue in the field, one that must be tackled to avoid repeating past mistakes and to ensure that AI is inclusive for all. This view is shared by other researchers in the field [1, 2].
> >
> > Additionally, FID and CLIP score measure similar concepts, but in different ways. CLIP score evaluates how closely a generated image matches a text prompt in the latent space of the CLIP network, while FID measures how close a generated image is to a ground truth image in the latent space of the Inception network. Both metrics are used to assess the quality of images generated by text-to-image models. We chose CLIP score over FID because FID requires a large number of image-text pairs to provide a reliable estimate, which would make a comprehensive experimental section impractical for our study.
> >
> > Finally, high-quality images do not necessarily imply an absence of bias in the training data. While extrapolation near the data distribution can lead to good predictions, it also provides insights into the boundaries of the training data and where potentially harmful predictions may emerge. Although our example in Figure 1 is close to the training data, it still reveals potential biases within the data.
> >
> > [1] Ferrara, Emilio. "Fairness and bias in artificial intelligence: A brief survey of sources, impacts, and mitigation strategies." Sci 6.1 (2023): 3.
> >
> > [2] Mehrabi, Ninareh, et al. "A survey on bias and fairness in machine learning." ACM computing surveys (CSUR) 54.6 (2021): 1-35.

---

### Official Review · Reviewer_S59P · 2024-11-01

**Soundness:** 1
**Presentation:** 2
**Contribution:** 1
**Rating:** 3
**Confidence:** 3

**Summary:**

This paper proposes to measure the epistemic uncertainty associated with text-to-image diffusion models by using a mixture-of-experts model constructed by using a collection of off-the-shelf diffusion models. The uncertainty is estimated via the aggregated variance across dimensions of a latent variable across ensemble members. Experimental results show alignment of the proposed metric with the CLIP score, indicating that a low uncertainty generally indicates high CLIP score.

**Strengths:**

+ This paper presents a simple method to quantify the uncertainty associated with the text-to-image generation process, potentially helping users decide whether or not a given generation may be accurate or not

+ The alignment with CLIP score seems to indicate that the proposed uncertainty measure seems to capture some important signal regarding the correctness of the generated image

**Weaknesses:**

One of the weaknesses of this paper is that it considers an **ill-motivated problem**. For example, Figure 1 is used as a motivating figure to demonstrate that text-to-image models may be biased. However, this figure does not seem to showcase any problems with these models, as the generated images for all prompts seem accurate. It is unclear how the observed descrepancy in the numerical uncertainty scores highlight any problem with the input-output behaviour of such models.

Additionally, the underlying problem is also **not well-posed mathematically**. In particular, while the paper aims to estimate the epistemic uncertainty, it does not specify the precise random variable of which the uncertainty is estimated. Because of this, it is unclear how one must interpret the numerical uncertainty estimate. While the method itself computes the variance across dimensions of a latent representation, it is unclear whether this fundamentally translates to uncertainty of any aspect of the input-output map. This results in the numerical uncertainty estimate itself being an **ill-motivated heuristic**, as without defining the random variable whose uncertainty is being estimated, it is hard to reason about whether the proposed metric is a good estimate or not.

Experimentally, the paper relies on the usage of the CLIP score to motivate its method, where it finds that the CLIP score generally aligns with the uncertainty metric. However, this fails to account for the fact that the **CLIP score may itself have issues with bias**, particularly that CLIP score may be higher for english text than finnish text due to the lack of finnish training data for training CLIP. This implies that experiments involving the CLIP score may not be useful in demonstrating correctness of the proposed metric.

Finally, I believe the paper **conflates ensembles and mixtures of experts**. Unless I am mistaken, the method in the paper seems to employ an ensemble, while the text claims usage of mixtures of experts.

Given these weaknesses, I am inclined to reject the paper at this time.

**Questions:**

I request the authors to address the weaknesses mentioned above. In particular, clarifying the mathematical and conceptual issues relating to the underlying random variable whose uncertainty is being estimated will help. In addition, some conceptual motivation regarding why the proposed uncertainty estimation metric is accurate will also improve this paper.

In addition, I request the authors to clarify how the proposed metric measures epistemic and not aleatoric uncertainty, as this is unclear in the draft.

(Minor Suggestion) In section 2, please specify the dimensions of all latent variables for clarity

---

> ### Author Response · Authors · 2024-11-19
>
> ## **Dear Reviewer,**
>
> Thank you for your insightful comments. Please find below our responses and the corresponding changes, which have also been highlighted in the provided PDF document in red.
>
>
> ### **Comment regarding Figure 1**
>
> Epistemic uncertainty provides practitioners with insights into inputs that were absent during training and areas where the model may be extrapolating. Accordingly, Figure 1 illustrates that images featuring Black male, White female, and Black female presidents exhibit higher uncertainty, suggesting that the model may not have encountered many such examples during training. This highlights bias within the training dataset and aligns with historical context, given the absence of female presidents and the presence of only one Black president. In general, diffusion models are capable of producing high-quality outputs regardless of whether the prompts have been encountered before or not during training.
>
> ### **Comment regarding not well-posed mathematically**
>
> Here is an intuitive explanation for our choice of estimator for epistemic uncertainty using the theory of Gaussian Processes. Each expert can be viewed as a sample from the posterior distribution of functions given an input y, denoted as $p(f (y)|y)$. By calculating the variance across these experts, we obtain the variance $\sigma^2$ of $p(f (y)|y)$, which serves as an estimate of epistemic uncertainty within the Gaussian Process framework. In general, other works have used the differences among ensemble components to represent epistemic uncertainty [1,2,3]. We have included this explanation, with additional mathematical rigor, in the appendix of our paper and highlighted it in the main text as follows:
>
> >The intuition behind this choice of epistemic uncertainty estimator is detailed in Appendix C.
>
> ### **Comment regarding CLIP score**
>
> Although CLIP score is known to exhibit bias [4], it remains a primary method for evaluating the quality of text-generated images, alongside the FID [5,6]. Both metrics rely on auxiliary networks (CLIP and the Inception network) for evaluation and are therefore both susceptible to bias. FID, however, requires a large number of samples for reliable estimates, whereas the CLIP score allows for more direct text-to-image alignment evaluation with fewer samples [7,8]. Given these factors, we prioritized the CLIP score for its suitability to our research objectives and its broad acceptance within related studies [5,7,8,9]. To further address potential concerns, we conducted an additional experiment evaluating EMoE’s performance using SSIM, detailed in Appendix B. The following text was added:
>
> >The CLIP score, despite its known biases (Chinchure et al., 2023), remains a widely-used method for evaluating the alignment between text prompts and generated images, alongside FID (Shi et al., 2020; Kumari et al., 2023). Both metrics, however, rely on auxiliary models (CLIP and Inception, respectively), making them susceptible to inherent biases. While FID requires a large number of samples for reliable estimation, the CLIP score facilitates a more direct assessment of text-to-image alignment with fewer samples (Kawar et al., 2023; Ho et al., 2022a). Considering these trade-offs, we prioritized the CLIP score due to its relevance to our research objectives and its broad acceptance in related studies.
> >
> >To further validate our findings and address any potential concerns related to metric biases, we conducted an additional experiment using the Structural Similarity Index (SSIM) as the evaluation metric. Unlike CLIP or FID, SSIM does not depend on any auxiliary models for its calculation, thereby mitigating the risk of bias. We computed SSIM between generated images and corresponding ground-truth images from the COCO dataset and analyzed the results for each uncertainty quartile. As shown in Table 5, EMoE effectively categorized prompts into the appropriate uncertainty quartiles based on model performance. This provides further evidence of EMoE’s efficacy in estimating uncertainty for MoE text-to-image models, highlighting its robustness across different evaluation metrics.
>
> You are correct that the CLIP score is generally lower for non-English prompts due to limited training data for CLIP in other languages. However, when evaluating the images generated from Finnish prompts, English prompts were then used to assess their CLIP score. We have added the following text for clarification:
>
> >Note that when evaluating the CLIP score for images generated from non-English prompts, the English version of the prompt was used for assessment. This was done to account for the fact that CLIP was primarily trained on English data.

---

> ### Author Response · Authors · 2024-11-19
>
> ### **Comment regarding MoE as an ensemble**
>
> We refer the reader to [10], which defines MoE (Mixture of Experts) as an ensemble method in Chapter 4.3. The ensemble methodology is defined as follows in [10]:
>
> >The main idea behind the ensemble methodology is to weigh several individual pattern classifiers and combine them, in order to obtain a classifier that outperforms all of them.
>
> Under this definition, MoE qualifies as an ensemble method, even if it selects the top expert and assigns zero weights to the remaining experts. EMoE extends this concept by disentangling the ensemble and treating each expert component as an independent model for estimating uncertainty.
>
> ### **References**
>
> [1] Yarin Gal, Riashat Islam, and Zoubin Ghahramani. Deep bayesian active learning with image data. In International Conference on Machine Learning, pp. 1183–1192. PMLR, 2017.
>
> [2] Stefan Depeweg, Jose-Miguel Hernandez-Lobato, Finale Doshi-Velez, and Steffen Udluft. Decomposition of uncertainty in bayesian deep learning for efficient and risk-sensitive learning. In International Conference on Machine Learning, pp. 1184–1193. PMLR, 2018.
>
> [3] Lucas Berry and David Meger. Normalizing flow ensembles for rich aleatoric and epistemic uncertainty modeling. Proceedings of the AAAI Conference on Artificial Intelligence, 37(6):6806–6814, 2023b.
>
> [4] Aditya Chinchure, Pushkar Shukla, Gaurav Bhatt, Kiri Salij, Kartik Hosanagar, Leonid Sigal, and Matthew Turk. Tibet: Identifying and evaluating biases in text-to-image generative models. arXiv preprint arXiv:2312.01261, 2023.
>
> [5] Zhan Shi, Xu Zhou, Xipeng Qiu, and Xiaodan Zhu. Improving image captioning with better use of captions. arXiv preprint arXiv:2006.11807, 2020.
>
> [6] Robin Rombach, Andreas Blattmann, Dominik Lorenz, Patrick Esser, and Bj¨orn Ommer. High-resolution image synthesis with latent diffusion models. In Proceedings of the IEEE/CVF Conference on Computer Vision and Pattern Recognition, pp. 10684–10695, 2022
>
> [7] Nupur Kumari, Bingliang Zhang, Richard Zhang, Eli Shechtman, and Jun-Yan Zhu. Multi-concept customization of text-to-image diffusion. In Proceedings of the IEEE/CVF Conference on Computer Vision and Pattern Recognition, pp. 1931–1941, 2023.
>
> [8] Bahjat Kawar, Shiran Zada, Oran Lang, Omer Tov, Huiwen Chang, Tali Dekel, Inbar Mosseri, and Michal Irani. Imagic: Text-based real image editing with diffusion models. In Proceedings of the IEEE/CVF Conference on Computer Vision and Pattern Recognition, pp. 6007–6017, 2023.
>
> [9] Jonathan Ho, Tim Salimans, Alexey Gritsenko, William Chan, Mohammad Norouzi, and David J Fleet. Video diffusion models. Advances in Neural Information Processing Systems, 35:8633–8646, 2022b.
>
> [10] Rokach, Lior. Ensemble learning: pattern classification using ensemble methods. 2019.

---

> > ### Author Response · Authors · 2024-11-25
> >
> > Dear Reviewer,
> >
> > As we approach the end of the author-reviewer discussion period, we wanted to kindly follow up regarding our revised manuscript and responses to your comments. We have addressed all the concerns you raised and incorporated the changes you suggested into the updated paper.
> >
> > If you have any remaining questions or additional feedback, please let us know so we can respond before the discussion period concludes. Your input is valuable to us, and we are committed to ensuring that our paper meets your expectations.
> >
> > We greatly appreciate your time and consideration.
> >
> > Best regards,
> >
> > The Authors

---

> > > ### Comment · Reviewer_S59P · 2024-11-26
> > >
> > > Thank you for your response!
> > >
> > > Regarding Figure 1, unfortunately, it is still unclear where the problem lies. I understand the argument that a higher uncertainty score indicates a model's tendency to extrapolate, but if the diffusion model works well despite the data imbalance and bias, where does the problem precisely lie? Any examples where a high uncertainty score systematically corresponds to **incorrect** images generated by the model would be useful to motivate the given method.
> > >
> > > Regarding the presented theory, I find it problematic in two aspects:
> > >
> > > (1) The theory considers variance of model outputs; whereas the method proposed considers variance along particular dimensions of a latent variable, averaged along dimensions. These are fundamentally distinct quantities.
> > >
> > > (2) The theory considers uncertainty in functions; however the paper computes variance across activations. Given that different parametric representations / activation maps can represent the same function, I believe the theory is misleading. Specifically, one can permute neurons / channels in successive layers of a neural network without changing its functionality. Thus, it is unclear how the proposed measure, which depends on model parameterization, is indicative of model functionality.
> > >
> > > Finally, the presented theory works for independent, identically distributed samples (i.e., models) from the posterior distribution, which are independently trained models forming an ensemble. I believe this aspect of the analysis applies to this paper because it is an ensemble and not a mixture of experts, as claimed in the paper. However, I consider this nomenclature issue (ensemble vs MoE) somewhat minor.
> > >
> > > Overall, due to the heuristic nature of the method and unclear evaluations and benefits, I am inclined to keep my score.

---

> > > > ### Author Response · Authors · 2024-12-04
> > > >
> > > > Dear Reviewer,
> > > >
> > > > Thank you for your feedback.
> > > >
> > > > ## Figure 1:
> > > >
> > > > Extrapolation, when close to the data distribution, can lead to good predictions. However, it still provides insight into the boundaries of the training data and highlights where potentially harmful predictions might arise. While our example is close to the training data, it still reveals potential biases within the data. Given that bias can have severe consequences for underrepresented populations, we believe the results presented in Figure 1 (and throughout the paper) are significant. Please note that bias in machine learning is a well-researched area [1, 2].
> > > >
> > > >
> > > > ## Theoretical Background:
> > > >
> > > > You are correct in noting that we are measuring uncertainty in the latent space rather than the output space. While our theoretical explanation is framed for the output space, in deep learning, it is often challenging to derive exact proofs, and researchers frequently rely on simplified assumptions. In this regard, our approach aligns with methods for estimating uncertainty in ensembles, as it is common for such methods to measure uncertainty in the latent spaces of networks [3, 4].
> > > >
> > > > ## MoE vs. Ensemble:
> > > >
> > > > We are still unclear about the specific concerns raised in your comments. A MoE can be seen as an ensemble where each expert is a component of that ensemble. In our case, each diffusion model was trained independently. Therefore, we consider the distance between experts (ensemble components) to provide an estimate of epistemic uncertainty.
> > > >
> > > > [1] Ferrara, Emilio. "Fairness and bias in artificial intelligence: A brief survey of sources, impacts, and mitigation strategies." Sci 6.1 (2023): 3.
> > > >
> > > > [2] Mehrabi, Ninareh, et al. "A survey on bias and fairness in machine learning." ACM computing surveys (CSUR) 54.6 (2021): 1-35.
> > > >
> > > > [3] Tomczak, Jakub, and Max Welling. "VAE with a VampPrior." International conference on artificial intelligence and statistics. PMLR, 2018.
> > > >
> > > > [4] Berry, Lucas, and David Meger. "Normalizing flow ensembles for rich aleatoric and epistemic uncertainty modeling." Proceedings of the AAAI Conference on Artificial Intelligence. Vol. 37. No. 6. 2023.

---

### Official Review · Reviewer_32yW · 2024-11-04

**Soundness:** 2
**Presentation:** 2
**Contribution:** 2
**Rating:** 5
**Confidence:** 4

**Summary:**

This paper introduces Epistemic Mixture of Experts (EMoE), a method for estimating epistemic uncertainty in text-to-image diffusion models by leveraging pre-trained models for zero-shot uncertainty estimation. EMoE leverages pre-trained networks for zero-shot uncertainty estimation, avoiding the need for additional training. In order to achieve this, they ensemble expert models from libraries such as HuggingFace during the diffusion process and compute the variance between them to give an uncertainty score. The method allows for early detection of uncertainty at the first U-Net layer, which could enable early stopping to save computational resources. To refine this score for final output, the authors additionally calculate variance after the U-Net mid-block. Experiments do show a correlation between uncertainty scores and CLIP scores, demonstrating its ability to measure uncertainty over previous work, DECU in and out of distribution. Additionally, they show experiments to support design choices such as ensemble size and estimating at the first layer. Overall, EMoE advances uncertainty estimation in diffusion models, but could use some additional clarifications in terms of expert selection. The authors mention that memory overhead is a potential concern, and I would agree that this could be a major limiting factor in the practicality of this approach.

**Strengths:**

- Their method estimates epistemic uncertainty without needing additional training as evidence by their correlation with CLIP scores
- The idea of separating the computational paths early to allow for early stopping is useful, saving potential computational cost
- They show results that this can perform well in out of distribution tasks

**Weaknesses:**

- Although they do not need to train their gating mechanism for their experiments, in a practical application, this would need to be done as it cannot account for more complex prompts
- More complex generation tasks would theoretically need more domain specific experts which could quickly add to computational overhead
- In figure 4 and table 1, the difference in clip score and word count between quartiles is very small (but does correlate with their score)
- The background section is almost 2 full pages. This definitely could have been filled with more results or ablations or methodology.

**Questions:**

- They mention that the quality and the diversity of their experts has a great effect on performance but have no analysis on how they selected their expert models.
- It might be useful to show an additional ablation that shows the amount of additional computational overhead their approach adds compared to base generation.

---

> ### Author Response · Authors · 2024-11-19
>
> ## **Dear Reviewer,**
>
> Thank you for your insightful comments. Please find below our responses and the corresponding changes, which have also been highlighted in the provided PDF document in red.
>
> ### **Comment regarding training of the gating network**
>
> You are correct in their assertion that the gating function of MoEs, as commonly used since their inception, requires training/fine-tuning. However a new gating method that does not require fine-tuning or re-training has emerged with LLMs and has become an established practice in the community (many models based on this approach are available on hugging face). The gating network used in our MoE approach is based on established community practices [1]. In our configuration, each expert is assigned to a positive and negative descriptor prompt. The positive descriptor emphasizes the expert’s strengths, while the negative descriptor highlights its limitations. During image generation, when a positive and negative prompt are provided, the system computes gating weights for each expert by measuring the distance between input prompts and corresponding descriptors in a latent space. This gating approach allows the re-combination of arbitrary expert models, that could be trained at differing times, on differing datasets, without fine-tuning. We have added a detailed explanation in Appendix D for further clarity.
>
> ### **Comment regarding computational burden of more experts**
>
> Using EMoE, it is possible to estimate uncertainty without generating an image for each expert. Instead, the variance of the midblock can be estimated at the initial reverse diffusion step, while employing the standard aggregation procedure from MoEs during subsequent image generation steps. This approach effectively reduces the computational burden associated with image generation. This enhanced version of EMoE results in only a minimal increase in computational time. The following addition to the paper describes this technique and quantifies the associated computational cost:
>
> >When running the SegMoE model in its standard mode, generating an image from one prompt takes an average of 3.58 seconds. In comparison, using EMoE typically requires an average of 12.32 seconds to generate four images from a single prompt. However, for scenarios where only one image per prompt is needed, EMoE’s output can be optimized by estimating epistemic uncertainty during the initial diffusion step, followed by standard MoE-based image generation. This optimized version of EMoE achieves an average generation time of 5.5 seconds. Table 10 provides further details.
>
> ### **Comment regarding the small deviation in CLIP score in Figure 4 and Table 1**
>
> The differences in CLIP scores in Figure 4 and Table 1 are relatively small because each expert has previously encountered the training data from Q1, Q2, Q3, and Q4. Consequently, if the experts are well-trained, they are expected to consistently produce high-quality images for all prompts within the training distribution. The capacity of EMoE to detect subtle variations in uncertainty, even with in-sample data, is a significant feature. To further clarify, we have added the following text:
>
> >Given that each expert has been trained on all data in the COCO dataset, EMoE’s ability to detect subtle differences in uncertainty on in-sample data is a notable feature.
>
> ### **Comment regarding background section**
>
> Given the complexity of the pipeline, we believe it is important to provide a thorough overview of uncertainty, UNets, and MoE. Accordingly, we have expanded the discussion on the methodology in the appendix.
>
> ### **Comment regarding diversity of experts**
>
> We provide an ablation study on the diversity of experts. In the Runway ML MoE, each expert represents a different version of Runway ML, illustrating an example of very similar models. EMoE demonstrates strong performance in this experiment (Figure 8d), showcasing EMoE's ability to detect uncertainty even among highly similar models. To clarify further, we have added the following text:
>
> >Additionally, this demonstrates that EMoE can detect uncertainty even within the context of very similar models.
>
> ### **References**
>
> [1] Goddard, Charles, et al. Arcee's MergeKit: A Toolkit for Merging Large Language Models. arXiv preprint arXiv:2403.13257 (2024).

---

> > ### Author Response · Authors · 2024-11-25
> >
> > Dear Reviewer,
> >
> > As we approach the end of the author-reviewer discussion period, we wanted to kindly follow up regarding our revised manuscript and responses to your comments. We have addressed all the concerns you raised and incorporated the changes you suggested into the updated paper.
> >
> > If you have any remaining questions or additional feedback, please let us know so we can respond before the discussion period concludes. Your input is valuable to us, and we are committed to ensuring that our paper meets your expectations.
> >
> > We greatly appreciate your time and consideration.
> >
> > Best regards,
> >
> > The Authors

---

> > > ### Comment · Reviewer_32yW · 2024-12-01
> > >
> > > Dear authors,
> > >
> > > Thank you for addressing my concerns. Due to the points other reviewers mentioned such as novelty of this work, I am not able to increase my score. However, given that my concerns were addressed in the rebuttal, I am keeping my score the same.
> > >
> > > Thanks.

---

> > > > ### Author Response · Authors · 2024-12-04
> > > >
> > > > Dear Reviewer,
> > > >
> > > > Thank you for your feedback. However, due to the general nature of your comments, it is difficult for us to fully address the remaining concerns.
> > > >
> > > > ## Novelty of EMoE:
> > > >
> > > > The differences between EMoE and previous methods [1, 2, 3] are mentioned in the paper. To reiterate:
> > > >
> > > > 1. **Zero-shot uncertainty estimation:** Our method is zero-shot, meaning no additional training is required to estimate uncertainty. In contrast, previous methods rely on additional training to estimate uncertainty. This feature allows EMoE to estimate uncertainty for models available on platforms like Hugging Face (over 30,000 models), without any extra training.
> > > > 2. **Text-to-image generation:** Unlike prior methods, which do not consider text-to-image generation, our method explicitly accounts for this use case. As a result, their approaches are not directly applicable to diffusion models, whereas EMoE is designed to handle these.
> > > > 3. **Epistemic uncertainty:** EMoE specifically estimates epistemic uncertainty, which enables bias detection and out-of-distribution detection—capabilities that previous methods do not offer.
> > > >
> > > > These unique characteristics make EMoE a novel solution in the field of uncertainty estimation for generative models.
> > > >
> > > > [1] Zhuobin Zheng, Chun Yuan, Xinrui Zhu, Zhihui Lin, Yangyang Cheng, Cheng Shi, and Jiahui Ye. Self-supervised mixture-of-experts by uncertainty estimation. In Proceedings of the AAAI Conference on Artificial Intelligence, volume 33, pp. 5933–5940, 2019.
> > > >
> > > > [2] Lucas Luttner. Training of neural networks with uncertain data, a mixture of experts approach. arXiv preprint arXiv:2312.08083, 2023.
> > > >
> > > > [3] Rongyu Zhang, Yulin Luo, Jiaming Liu, Huanrui Yang, Zhen Dong, Denis Gudovskiy, Tomoyuki Okuno, Yohei Nakata, Kurt Keutzer, Yuan Du, et al. Efficient deweahter mixture-of-experts with uncertainty-aware feature-wise linear modulation. In Proceedings of the AAAI Conference on Artificial Intelligence, volume 38, pp. 16812–16820, 2024.

---

### Official Review · Reviewer_PQNu · 2024-11-07

**Soundness:** 2
**Presentation:** 2
**Contribution:** 2
**Rating:** 5
**Confidence:** 4

**Summary:**

This paper introduces an Epistemic Mixture of Experts (EMoE), a framework that efficiently estimates epistemic uncertainty without additional training by leveraging pre-trained networks. EMoE operates within the latent space in the diffusion process, capturing uncertainty early in the denoising phase. The authors demonstrate the results on the COCO dataset, where the alignment of uncertainty estimates with image quality is chosen as the evaluation criteria. This work also claims to identify biases, such as under-represented languages and regions.

**Strengths:**

- The authors address a significant and well-explored challenge: uncertainty quantification in text-to-image diffusion models.
- The proposed use of EMoE for uncertainty quantification is straightforward and offers clear interpretability.
- The authors present a methodology for integrating the MOE network into text-to-image diffusion models to quantify epistemic uncertainty.
- The literature review is comprehensive, effectively situating the work within the context of existing research.

**Weaknesses:**

- The core component of this work is the Mixture of Experts (MOE) network, a well-established method for uncertainty quantification, as demonstrated in previous studies [1,2,3]. This reliance on a pre-existing approach may limit the perceived novelty of this work, as it may appear to simply adapt a plug-and-play method to diffusion models without substantial innovation.

- The authors rely solely on the CLIP score to evaluate their model’s performance, which may raise issues since CLIP has known biases. For example, CLIP often correlates “man” more strongly with “woman” than with “boy,” suggesting potential misalignment with accurate demographic representation. Refer to [4] for additional details on CLIP’s limitations in bias.

- Epistemic uncertainty, a primary focus of this work, is also highly relevant to the concept of hallucinations in generative models, but this connection is absent in the manuscript. For more on this topic, see [5].

- The motivating examples (Figure 1) illustrate the link between epistemic uncertainty and concerns of bias and fairness. However, the paper lacks a thorough discussion of how EMoE addresses bias and fairness within text-to-image models, a critical aspect of the model's real-world implications.

[1] Self-Supervised Mixture-of-Experts by Uncertainty Estimation

[2] Training of Neural Networks with Uncertain Data: A Mixture of Experts Approach

[3] Efficient Deweather Mixture-of-Experts with Uncertainty-aware Feature-wise Linear Modulation

[4] TIBET: Identifying and Evaluating Biases in Text-to-Image Generative Models

[5] Understanding Hallucinations in Diffusion Models through Mode Interpolation

**Questions:**

- Since the CLIP score has known biases, particularly with demographic representation, why did you choose it as the sole performance metric? Could you discuss any potential impacts of these biases on your evaluation and whether you considered any additional metrics to provide a more comprehensive assessment?

-  Epistemic uncertainty is closely linked to hallucinations in generative models, yet this is not addressed in the manuscript. Could you explain how EMoE might capture or identify hallucinations in generated images, and why this aspect was not included in the current analysis?

-  A connection could be made with the hallucination metric proposed in [5], allowing for a comparison. This might serve as an alternative to the CLIP metric.

- With reference to Figure 1 there may exist a relationship between epistemic uncertainty and issues of bias and fairness. Could you expand on how EMoE addresses these concerns, and whether additional analysis on bias and fairness in generated content was considered?

- Since EMoE aims to provide interpretable uncertainty estimates, could you discuss how practitioners could use these estimates to improve model reliability or detect potential biases? Are there any specific guidelines for interpreting and acting on the model’s uncertainty scores?


[5] Understanding Hallucinations in Diffusion Models through Mode Interpolation

---

> ### Author Response · Authors · 2024-11-19
>
> ## **Dear Reviewer,**
>
> Thank you for your insightful comments. Please find below our responses and the corresponding changes, which have also been highlighted in the provided PDF document in red.
>
> ### **Comments Regarding Uncertainty in Previous MoE Papers**
>
> We appreciate you bringing up the missing related work. We would like to clarify and expand upon the treatment of uncertainty in these references:
>
> - [1] does not estimate epistemic uncertainty; instead, the uncertainty estimation is isolated to the critic component, where the variance of a Gaussian distribution is estimated using negative log-likelihood as the loss function. Importantly, the MoE utilized in [1] is separate from the critic, meaning that the uncertainty estimation does not originate from a MoE component. Furthermore, their approach necessitates training the entire pipeline, precluding zero-shot capabilities.
>
> - [2] specifically addresses aleatoric uncertainty in MoE models. Our work, in contrast, focuses on epistemic uncertainty, offering a distinct contribution in this space.
>
> - [3] does not report any uncertainty estimates. Instead, it leverages uncertainty to assign experts more efficiently within MoE models. Additionally, it is noteworthy that [3] requires training its uncertainty-aware routing (UaR), whereas our proposed EMoE approach estimates uncertainty without requiring any training.
>
> It is important to highlight that none of these approaches—[1], [2], or [3]—estimate epistemic uncertainty and all require training from scratch. Moreover, none of these works address text-to-image generation tasks. Specifically, [1] focuses on regression and classification problems, [2] evaluates on reinforcement learning (RL) tasks, and [3] concentrates on the removal of weather artifacts from images.To address these distinctions, we have included the following addition in the related works section:
>
> >Further, while previous methods have integrated uncertainty into model pipelines using MoE (Zheng et al., 2019; Luttner, 2023; Zhang et al., 2024), these approaches neither address epistemic uncertainty nor consider text-to-image generation tasks and are not applicable in a zero-shot manner.
>
> ### **Comments regarding CLIP score**
>
> Although CLIP score is known to exhibit bias [4], it remains a primary method for evaluating the quality of text-generated images, alongside the FID [5,6]. Both metrics rely on auxiliary networks (CLIP and the Inception network) for evaluation and are therefore both susceptible to bias. FID, however, requires a large number of samples for reliable estimates, whereas the CLIP score allows for more direct text-to-image alignment evaluation with fewer samples [6,7]. Given these factors, we prioritized the CLIP score for its suitability to our research objectives and its broad acceptance within related studies [4,6,7,8]. To further address potential concerns, we conducted an additional experiment evaluating EMoE’s performance using SSIM, detailed in Appendix B. The following text was added:
>
> >The CLIP score, despite its known biases (Chinchure et al., 2023), remains a widely-used method for evaluating the alignment between text prompts and generated images, alongside FID (Shi et al., 2020; Kumari et al., 2023). Both metrics, however, rely on auxiliary models (CLIP and Inception, respectively), making them susceptible to inherent biases. While FID requires a large number of samples for reliable estimation, the CLIP score facilitates a more direct assessment of text-to-image alignment with fewer samples (Kawar et al., 2023; Ho et al., 2022a). Considering these trade-offs, we prioritized the CLIP score due to its relevance to our research objectives and its broad acceptance in related studies.
> >
> >To further validate our findings and address any potential concerns related to metric biases, we conducted an additional experiment using the Structural Similarity Index (SSIM) as the evaluation metric. Unlike CLIP or FID, SSIM does not depend on any auxiliary models for its calculation, thereby mitigating the risk of bias. We computed SSIM between generated images and corresponding ground-truth images from the COCO dataset and analyzed the results for each uncertainty quartile. As shown in Table 5, EMoE effectively categorized prompts into the appropriate uncertainty quartiles based on model performance. This provides further evidence of EMoE’s efficacy in estimating uncertainty for MoE text-to-image models, highlighting its robustness across different evaluation metrics.

---

> ### Author Response · Authors · 2024-11-19
>
> ### **Comments on hallucinations**
>
> We address the connection between EMoE and the concept of hallucinations in generative models in the related works section, lines 510-513. The concept of hallucinations in such models is still in its early stages of development, as evidenced by the reference you provided [10], published in the summer of 2024 and still undergoing formal definition. Our work does not aim to formally define hallucinations; rather, we have offered a brief commentary on the topic. Additionally, we do not attempt to capture hallucinations such as in [10] since we did not train the networks ourselves and lack detailed knowledge of the training data, making it impossible to accurately determine what constitutes a hallucinated response. Therefore, it does not make sense to use [10] as a baseline given our experimental setup.
>
> ### **Comments on Figure 1 and Detecting Biases and Fairness**
>
> We acknowledge that the paper does not specifically address racial and gender biases as illustrated in Figure 1. However, we present multiple experiments demonstrating how EMoE can be employed to detect language biases in MoE diffusion models, as discussed in Sections 4.2 and 4.3. This highlights which languages are underrepresented by these models, showcasing their implications for fairness. Practitioners may leverage EMoE similarly, as shown in Sections 4.2 and 4.3, to fine tune diffusion models on underrepresented languages. It is worth noting that Figure 1 was intended as an accessible and illustrative motivating example. We have added the following text to the paper to clarify this:
>
> >This section and Section 4.2 illustrate the model’s bias toward certain languages and reveal its unfairness toward non-European languages. This demonstrates how EMoE can be utilized to detect biases and identify the data necessary for training to mitigate these issues.
>
> ### **References**
>
> [1] Zhuobin Zheng, Chun Yuan, Xinrui Zhu, Zhihui Lin, Yangyang Cheng, Cheng Shi, and Jiahui Ye. Self-supervised mixture-of-experts by uncertainty estimation. In Proceedings of the AAAI Conference on Artificial Intelligence, volume 33, pp. 5933–5940, 2019.
>
> [2] Lucas Luttner. Training of neural networks with uncertain data, a mixture of experts approach. arXiv preprint arXiv:2312.08083, 2023.
>
> [3] Rongyu Zhang, Yulin Luo, Jiaming Liu, Huanrui Yang, Zhen Dong, Denis Gudovskiy, Tomoyuki Okuno, Yohei Nakata, Kurt Keutzer, Yuan Du, et al. Efficient deweahter mixture-of-experts with uncertainty-aware feature-wise linear modulation. In Proceedings of the AAAI Conference on Artificial Intelligence, volume 38, pp. 16812–16820, 2024.
>
> [4] Aditya Chinchure, Pushkar Shukla, Gaurav Bhatt, Kiri Salij, Kartik Hosanagar, Leonid Sigal, and Matthew Turk. Tibet: Identifying and evaluating biases in text-to-image generative models. arXiv preprint arXiv:2312.01261, 2023.
>
> [5] Zhan Shi, Xu Zhou, Xipeng Qiu, and Xiaodan Zhu. Improving image captioning with better use of captions. arXiv preprint arXiv:2006.11807, 2020.
>
> [6] Robin Rombach, Andreas Blattmann, Dominik Lorenz, Patrick Esser, and Bj¨orn Ommer. High-resolution image synthesis with latent diffusion models. In Proceedings of the IEEE/CVF Conference on Computer Vision and Pattern Recognition, pp. 10684–10695, 2022
>
> [7] Nupur Kumari, Bingliang Zhang, Richard Zhang, Eli Shechtman, and Jun-Yan Zhu. Multi-concept customization of text-to-image diffusion. In Proceedings of the IEEE/CVF Conference on Computer Vision and Pattern Recognition, pp. 1931–1941, 2023.
>
> [8] Bahjat Kawar, Shiran Zada, Oran Lang, Omer Tov, Huiwen Chang, Tali Dekel, Inbar Mosseri, and Michal Irani. Imagic: Text-based real image editing with diffusion models. In Proceedings of the IEEE/CVF Conference on Computer Vision and Pattern Recognition, pp. 6007–6017, 2023.
>
> [9] Jonathan Ho, Tim Salimans, Alexey Gritsenko, William Chan, Mohammad Norouzi, and David J Fleet. Video diffusion models. Advances in Neural Information Processing Systems, 35:8633–8646, 2022b.
>
> [10] Aithal, Sumukh K., et al. Understanding Hallucinations in Diffusion Models through Mode Interpolation. arXiv preprint arXiv:2406.09358 2024.

---

> > ### Author Response · Authors · 2024-11-25
> >
> > Dear Reviewer,
> >
> > As we approach the end of the author-reviewer discussion period, we wanted to kindly follow up regarding our revised manuscript and responses to your comments. We have addressed all the concerns you raised and incorporated the changes you suggested into the updated paper.
> >
> > If you have any remaining questions or additional feedback, please let us know so we can respond before the discussion period concludes. Your input is valuable to us, and we are committed to ensuring that our paper meets your expectations.
> >
> > We greatly appreciate your time and consideration.
> >
> > Best regards,
> >
> > The Authors

---

> > > ### Comment · Reviewer_PQNu · 2024-11-26
> > >
> > > Thank you for addressing my questions. While the authors have resolved many of my concerns, some key issues remain:
> > >
> > > - **Replacing the CLIP score with the SSIM score:** SSIM relies solely on visual information and does not account for contextual similarities. While CLIP is known to be biased, SSIM lacks semantic and contextual measures. I acknowledge the authors’ point that these metrics have inherent limitations, which can serve as inductive biases within the given framework.
> > >
> > > - **Novelty of MoE in this work:** Although the authors have clarified the differences between EMoE and prior works utilizing MoE for uncertainty estimation, it is unclear what specifically makes the MoE approach in this work a novel solution.
> > >
> > > - **Discussion on hallucination:** I had suggested addressing hallucination to strengthen the proposed work, as it could illustrate the usefulness of EMoE. Since hallucination remains a significant and relatively unexplored topic, any connection between EMoE and hallucination would enhance the impact of the manuscript. However, I acknowledge that this discussion may fall outside the scope of the current work, and its absence will not affect my evaluation.
> > >
> > > Overall, my review remains borderline for the current version of the manuscript, and I will retain my score.

---

> > > > ### Author Response · Authors · 2024-12-04
> > > >
> > > > Dear Reviewer,
> > > >
> > > > Thank you for your feedback.
> > > >
> > > > ## Regarding the Replacement of CLIP Score with SSIM:
> > > >
> > > > We would like to clarify that we did not replace the CLIP score with SSIM; instead, we added an additional experiment that includes SSIM in the appendix. None of the original experiments in the paper were removed. You are correct that SSIM lacks a contextual basis, and for this reason, it is not our primary evaluation metric. Your comments suggest the need for a metric that is both unbiased and capable of evaluating text-to-image models with respect to semantic and contextual information. Unfortunately, no such comprehensive metric currently exists. That said, CLIP score is a well-established and widely accepted metric in the text-to-image community [1,2,3,4,5]. Dismissing the use of CLIP score in our experiments would go against the best practices in the field.
> > > >
> > > > ## Novelty of EMoE:
> > > >
> > > > The reviewer states:
> > > >
> > > > Although the authors have clarified the differences between EMoE and prior works utilizing MoE for uncertainty estimation, it is unclear what specifically makes the MoE approach in this work a novel solution.
> > > >
> > > > This suggests that the reviewer has understood the differences between EMoE and previous methods [6, 7, 8]. To reiterate:
> > > >
> > > > 1. **Zero-shot uncertainty estimation:** Our method is zero-shot, meaning no additional training is required to estimate uncertainty. In contrast, previous methods rely on additional training to estimate uncertainty. This feature allows EMoE to estimate uncertainty for models available on platforms like Hugging Face (over 30,000 models), without any extra training.
> > > > 2. **Text-to-image generation:** Unlike prior methods, which do not consider text-to-image generation, our method explicitly accounts for this use case. As a result, their approaches are not directly applicable to diffusion models, whereas EMoE is designed to handle these.
> > > > 3. **Epistemic uncertainty:** EMoE specifically estimates epistemic uncertainty, which enables bias detection and out-of-distribution detection—capabilities that previous methods do not offer.
> > > >
> > > > These unique characteristics make EMoE a novel solution in the field of uncertainty estimation for generative models.
> > > >
> > > > ## Hallucinations:
> > > >
> > > > We do not provide any experiments on hallucinations in this work, but we do include comments regarding EMoE’s potential for hallucination detection. Bias remains a crucial problem for generative models and can have detrimental long-term effects on underrepresented populations in machine learning [9, 10]. Given these concerns, we focused our efforts on exploring bias detection in text-to-image MoE models, as we deemed this a sufficient and impactful direction.
> > > >
> > > >
> > > > [1] Saharia, Chitwan, et al. "Photorealistic text-to-image diffusion models with deep language understanding." Advances in neural information processing systems 35 (2022): 36479-36494.
> > > >
> > > > [2] Kang, Minguk, et al. "Scaling up gans for text-to-image synthesis." Proceedings of the IEEE/CVF Conference on Computer Vision and Pattern Recognition. 2023.
> > > >
> > > > [3] Kumari, Nupur, et al. "Multi-concept customization of text-to-image diffusion." Proceedings of the IEEE/CVF Conference on Computer Vision and Pattern Recognition. 2023.
> > > >
> > > > [4] Ho, Jonathan, et al. "Imagen video: High definition video generation with diffusion models." arXiv preprint arXiv:2210.02303 (2022)..
> > > >
> > > > [5] Kawar, Bahjat, et al. "Imagic: Text-based real image editing with diffusion models." Proceedings of the IEEE/CVF Conference on Computer Vision and Pattern Recognition. 2023.
> > > >
> > > > [6] Zhuobin Zheng, Chun Yuan, Xinrui Zhu, Zhihui Lin, Yangyang Cheng, Cheng Shi, and Jiahui Ye. Self-supervised mixture-of-experts by uncertainty estimation. In Proceedings of the AAAI Conference on Artificial Intelligence, volume 33, pp. 5933–5940, 2019.
> > > >
> > > > [7] Lucas Luttner. Training of neural networks with uncertain data, a mixture of experts approach. arXiv preprint arXiv:2312.08083, 2023.
> > > >
> > > > [8] Rongyu Zhang, Yulin Luo, Jiaming Liu, Huanrui Yang, Zhen Dong, Denis Gudovskiy, Tomoyuki Okuno, Yohei Nakata, Kurt Keutzer, Yuan Du, et al. Efficient deweahter mixture-of-experts with uncertainty-aware feature-wise linear modulation. In Proceedings of the AAAI Conference on Artificial Intelligence, volume 38, pp. 16812–16820, 2024.
> > > >
> > > > [9] Ferrara, Emilio. "Fairness and bias in artificial intelligence: A brief survey of sources, impacts, and mitigation strategies." Sci 6.1 (2023): 3.
> > > >
> > > > [10] Mehrabi, Ninareh, et al. "A survey on bias and fairness in machine learning." ACM computing surveys (CSUR) 54.6 (2021): 1-35.

---

### Author Response · Authors · 2024-11-19

### **References**

[1]Goddard, Charles, et al. Arcee's MergeKit: A Toolkit for Merging Large Language Models. arXiv preprint arXiv:2403.13257 (2024).

[2] Aditya Chinchure, Pushkar Shukla, Gaurav Bhatt, Kiri Salij, Kartik Hosanagar, Leonid Sigal, and Matthew Turk. Tibet: Identifying and evaluating biases in text-to-image generative models. arXiv preprint arXiv:2312.01261, 2023.

[3] Zhan Shi, Xu Zhou, Xipeng Qiu, and Xiaodan Zhu. Improving image captioning with better use of captions. arXiv preprint arXiv:2006.11807, 2020.

[4] Robin Rombach, Andreas Blattmann, Dominik Lorenz, Patrick Esser, and Bj¨orn Ommer. High-resolution image synthesis with latent diffusion models. In Proceedings of the IEEE/CVF Conference on Computer Vision and Pattern Recognition, pp. 10684–10695, 2022

[5] Nupur Kumari, Bingliang Zhang, Richard Zhang, Eli Shechtman, and Jun-Yan Zhu. Multi-concept customization of text-to-image diffusion. In Proceedings of the IEEE/CVF Conference on Computer Vision and Pattern Recognition, pp. 1931–1941, 2023.

[6] Bahjat Kawar, Shiran Zada, Oran Lang, Omer Tov, Huiwen Chang, Tali Dekel, Inbar Mosseri, and Michal Irani. Imagic: Text-based real image editing with diffusion models. In Proceedings of the IEEE/CVF Conference on Computer Vision and Pattern Recognition, pp. 6007–6017, 2023.

[7] Jonathan Ho, Tim Salimans, Alexey Gritsenko, William Chan, Mohammad Norouzi, and David J Fleet. Video diffusion models. Advances in Neural Information Processing Systems, 35:8633–8646, 2022b.

---

### Author Response · Authors · 2024-11-19

## **Meta Rebuttal**

We thank all reviewers for their insightful comments. All highlighted changes have been incorporated into the paper. Here, we address some common concerns raised across multiple reviews.

### **Figure 1**

Although our paper does not directly focus on racial and gender biases highlighted in Figure 1, we provide multiple experiments demonstrating how EMoE can detect language biases in MoE diffusion models, as discussed in Sections 4.2 and 4.3. This reveals underrepresentation of certain languages within training datasets, with significant implications for fairness. Practitioners can utilize EMoE, as demonstrated in these sections, to isolate which regions of the input space need to be fine-tuned to better represent underrepresented languages. Figure 1 is an example of how our method can be used to highlight the bias and fairness present in the datasets commonly used to train these models. We have added the following clarification to the paper:

>This section and Section 4.2 illustrate the model’s bias toward certain languages and reveal its unfairness toward non-European languages. This demonstrates how EMoE can be utilized to detect biases and identify the data necessary for training to mitigate these issues.

### **How the Gate Network Works**

The reviewers are correct in their assertion that the gating function of MoEs, as commonly used since their inception, requires training/fine-tuning. However a new gating method that does not require fine-tuning or re-training has emerged with LLMs and has become an established practice in the community (many models based on this approach are available on hugging face). The gating network used in our MoE approach is based on established community practices [1]. In our configuration, each expert is assigned to a positive and negative descriptor prompt. The positive descriptor emphasizes the expert’s strengths, while the negative descriptor highlights its limitations. During image generation, when a positive and negative prompt are provided, the system computes gating weights for each expert by measuring the distance between input prompts and corresponding descriptors in a latent space. This gating approach allows the re-combination of arbitrary expert models, that could be trained at differing times, on differing datasets, without fine-tuning. We have added a detailed explanation in Appendix D for further clarity.

### **Justification of CLIP score**

Although CLIP score is known to exhibit bias [2], it remains a primary method for evaluating the quality of text-generated images, alongside the FID [3,4]. Both metrics rely on auxiliary networks (CLIP and the Inception network) for evaluation and are therefore both susceptible to bias. FID, however, requires a large number of samples for reliable estimates, whereas the CLIP score allows for more direct text-to-image alignment evaluation with fewer samples [5,6]. Given these factors, we prioritized the CLIP score for its suitability to our research objectives and its broad acceptance within related studies [3,5,6,7]. To further address potential concerns, we conducted an additional experiment evaluating EMoE’s performance using SSIM, detailed in Appendix B. The following text was added:

>The CLIP score, despite its known biases (Chinchure et al., 2023), remains a widely-used method for evaluating the alignment between text prompts and generated images, alongside FID (Shi et al., 2020; Kumari et al., 2023). Both metrics, however, rely on auxiliary models
(CLIP and Inception, respectively), making them susceptible to inherent biases. While FID requires a large number of samples for reliable estimation, the CLIP score facilitates a more direct assessment of text-to-image alignment with fewer samples (Kawar et al., 2023; Ho et al., 2022a). Considering these trade-offs, we prioritized the CLIP score due to its relevance to our research objectives and its broad acceptance in related studies.
>
>To further validate our findings and address any potential concerns related to metric biases, we conducted an additional experiment using the Structural Similarity Index (SSIM) as the evaluation metric. Unlike CLIP or FID, SSIM does not depend on any auxiliary models for its calculation, thereby mitigating the risk of bias. We computed SSIM between generated images and corresponding ground-truth images from the COCO dataset and analyzed the results for each uncertainty quartile. As shown in Table 5, EMoE effectively categorized prompts into the appropriate uncertainty quartiles based on model performance. This provides further evidence of EMoE’s efficacy in estimating uncertainty for MoE text-to-image models, highlighting its robustness across different evaluation metrics.

---

### Comment · Area_Chair_EFbC · 2024-11-25
**The author-reviewer discussion period is ending soon**

Dear reviewers,

If you haven’t done so already, please engage in the discussion as soon as possible. Specifically, please acknowledge that you have thoroughly reviewed the authors' rebuttal and indicate whether your concerns have been adequately addressed. Your input during this critical phase is essential—not only for the authors but also for your fellow reviewers and the Area Chair—to ensure a fair evaluation.

Best wishes,
AC

---

### Meta-Review · Area_Chair_EFbC · 2024-12-19

**Metareview:**

This paper introduces a framework for estimating epistemic uncertainty in text-to-image diffusion models using a mixture of experts. However, the reviewers expressed concerns about the novelty of the method and found many heuristic aspects difficult to justify or clearly explain. Despite attempts to address these issues during the rebuttal period, the responses were not entirely successful. I encourage the authors to further develop the method by strengthening its theoretical foundation or addressing the reviewers' concerns more clearly, such as the biases associated with the use of CLIP scores.

**Additional Comments On Reviewer Discussion:**

The reviewers raised concerns about the motivation and justification behind the proposed methods, and the authors were unable to adequately address these issues during the rebuttal period. The consensus remained unchanged during the reviewer discussion phase.

---

### Decision · Program_Chairs · 2025-01-22

Reject